# Robust Multi-Mode Synchronization of Chaotic Fractional Order Systems in the Presence of Disturbance, Time Delay and Uncertainty with Application in Secure Communications

**Ali Akbar Kekha Javan [1], Assef Zare [1,\*], Roohallah Alizadehsani [2] and Saeed Balochian [1]**

1    Department of Electrical Engineering, Gonabad Branch, Islamic Azad University, Gonabad 6518115743, Iran; kikha_akbar@yahoo.com (A.A.K.J.); saeed.balochian@yahoo.com (S.B.)
2    Institute for Intelligent Systems Research and Innovation (I.I.S.R.I.), Deakin University, Geelong, VIC 3216, Australia; r.alizadehsani@deakin.edu.au
*    Correspondence: assefzare@gmail.com

**Abstract:** This paper investigates the robust adaptive synchronization of multi-mode fractional-order chaotic systems (MMFOCS). To that end, synchronization was performed with unknown parameters, unknown time delays, the presence of disturbance, and uncertainty with the unknown boundary. The convergence of the synchronization error to zero was guaranteed using the Lyapunov function. Additionally, the control rules were extracted as explicit continuous functions. An image encryption approach was proposed based on maps with time-dependent coding for secure communication. The simulations indicated the effectiveness of the proposed design regarding the suitability of the parameters, the convergence of errors, and robustness. Subsequently, the presented method was applied to fractional-order Chen systems and was encrypted using the chaotic masking of different benchmark images. The results indicated the desirable performance of the proposed method in encrypting the benchmark images.

**Keywords:** multi-mode synchronization; adaptive-robust synchronization; fractional order hyperchaotic system; lyapunov stability; secure communication

## 1. Introduction

Today, many people use digital communications in daily activities, changing this field into one of the most special human needs [1–3]. Additionally, digital communications facilitate commerce in different industrial and medical contexts [4,5]. In that regard, information security is vital. The digital communications system includes a transmitter, communication channel, and receiver. Information security in communication channels is of great importance. The particular importance of information security in digital communications has led to extensive studies in this domain, and scientists have presented various methods [6–9]. Encryption is one of the most important information security contexts in digital communications and includes multiple methods such as chaos theory techniques [10]. Numerous synchronization techniques for various chaotic systems have been proposed based on control techniques. Generally, chaotic systems fall into standard and fractional order types, and each is analyzed in many synchronization systems. Control methods applied in synchronization incorporate linear and nonlinear [11,12], fuzzy [13], active [14,15], backstepping [16,17], sliding mode control (SMC) [18,19], and adaptive control [20]. Lai used various methods such as designing a new memristor that is flux-controlled [21], a Lü chaos system with coexisting attractors and a nonlinear controller [22], and a new encryption algorithm performed on hyper chaotic neurons [23] to enhance image encryption security.

The idea of this paper is to employ SMC alongside three chaotic systems. To further increase efficiency, the parameters of the chaotic systems that are unknown are introduced. The authors in [24] investigated Hopf bifurcation and amplitude controls in the chaotic

Toda jerk oscillator system with analysis, circuit realization, and combined synchronization in the fractional-order form. In [25], the multi-switching synchronization of several chaotic systems was examined. The proposed Lorenz and Chen chaotic systems were adopted and explained three-dimensionally. However, in relevant studies, it has been assumed that their parameters are known. In [26], the combination synchronization of a new fractional-order chaotic system with two stable node-foci was studied. In [27], dual-phase and dual anti-phase synchronization of FOCS in real and complex variables with uncertainties was investigated. The effect of disturbance is not considered in this paper. The authors in [28] studied the combination synchronization of three fractional-order time-delayed chaotic systems without disturbance and uncertainty. The authors in [29] surveyed multi-switching combination-combination synchronization of unknown FOCS, in which three-dimensional systems were exploited, and the effect of disturbance was not studied. Various studies deal with system synchronization applications in secure communication: Khan and Nigar (2020) used sliding-mode disturbance observer control based on adaptive hybrid projective combination synchronization in fractional-order systems [30]. An adaptive-robust control for multi-state synchronization of chaotic systems with unknown and time-varying delay was studied by Kekha javan [31].

A chaotic system with an exponential term was considered in [32] to study its synchronization. It seeks the dynamics of approximation of exponential chaotic systems. Moreover, combination synchronization was used to control this system. Afterward, its application was exhibited in secure communication. An application of uncertain chaotic system adaptive synchronization was studied in secure communication systems in [33]. Khan et al., (2019) proposed a secure communication design through novel fractional chaotic system synchronization [34]. A finite-time terminal auxiliary observer was used for fractional-order chaotic systems with application in secure communication [35]. The authors in [36] used secure communication via projective synchronization of the fractional matrix. In [37], a secure communication design was used based on the modified hybrid projective synchronization of fractional hyper-chaotic systems. A novel acoustic encryption method with finite-time synchronization of the fractional-order hyper-chaotic system was proposed [38]. Vaseghi et al. (2021) applied finite-time synchronization in chaotic systems to encrypt medical images [39].

Previous research indicates the gap regarding the multi-mode synchronization of fractional-order hyper-chaotic systems with unknown and variable parameters using an adaptive-robust controller. Accordingly, this paper aims to demonstrate a multi-mode synchronization scheme for fractional-order hyper-chaotic systems with parametric uncertainty and disturbances. The boundaries of these uncertainties and disturbances are estimated by applying adaptive rules. Furthermore, its detrimental effect on the performance of synchronization control law is significantly reduced. The proposed novel secure communication method uses chaotic masking with a new modulation function for benchmark image encryption. Information entropy, signal-to-noise ratio, histogram, and similar criteria were used to analyze the proposed method.

The main contributions of the present study are as follows:

(1) Synchronization, despite the stepwise changes of system parameters.
(2) Determination of control rule as an explicit and continuous function that prohibits the manifestation of chattering phenomenon.
(3) Synchronization is independent of the type of chaotic system.
(4) Determination of several adaptive rules such that the system stability is guaranteed and the convergence of synchronization errors and estimating errors of disturbance and uncertainty boundaries converge to zero.
(5) A novel secure communication design was considered the modulation function for chaotic masking.

To evaluate the performance of the proposed scheme, simulations were carried out on three fractional-order hyper-chaotic Chen systems with unknown parameters and external

disturbances. Their results indicated the adequacy and robustness of the proposed adaptive controller scheme.

The next part of this paper presents the formulation of the proposed method. It contains the fractional-order systems and adaptive synchronization of multiple FOCS with unknown parameters expressed in two transmission and circular multi-mode synchronization. This issue was also tested in the presence of disturbance. Section 4 presents the simulation results. The standard benchmark images were simulated and evaluated with the statistic criterion. The efficiency of the method was evaluated by histogram analysis, correlation analysis, number of pixel change rate (NPCR), unified average changing intensity (UACI), peak signal to noise ratio (PSNR), and information entropy. The experiment results indicated the proposed encryption method's ability to uplift security and performance. Section 6 presents the discussion of the proposed method, followed by the conclusion.

## 2. Problem Formulation

This section discusses the proposed scheme of the paper. First, fractional-order systems are presented. Subsequently, the synchronization of chaotic systems of multiple transmission fractional order is described. Finally, the multiple circular synchronizations are discussed. In both cases, adaptive rules and controllers are designed using the adaptive control method. Two examples are given to demonstrate the efficiency and performance of the proposed method.

### 2.1. Basic Definitions
Fractional-Order Derivative

Due to simple implementation and high performance, several numerical definitions have been proposed to solve fractional differential equations [40]. This paper uses the definition of Caputo, whose fractional derivative is as follows [40]:

$$D^q f(x) = I^{m-q} h^{(m)}(x), \ q > 0 \tag{1}$$

where $h^{(m)}$ represents the derivative of mth order of $h(x)$, $m = [q]$ is the first integer that is less than $q$, and the Riemann-Lewil integral operator with order $q$ of function $g(x)$ is described as follows [40]:

$$I^q g(x) = \frac{1}{\Gamma(q)} \int_0^x (x - t)^{q-1} g(t) dt, \ q > 0 \tag{2}$$

where $\Gamma(q)$ is the gamma function and the $D^q$ operator is called the Caputo fractional operator of $q$ order.

Stability analysis of fractional-order systems by Lyapunov's direct method and determining the necessary and sufficient conditions guaranteeing stability with the Mittag-Leffler concept [41] and stability analysis based on convex of the Lyapunov functions [40] for nonlinear systems are demonstrated.

**Lemma 1.** [40]: *Suppose $h(t) \in R$ is a continuous and derivable function. Then, for $t \geq t_0$, we have:*

$$D^q h^2(t) \leq 2h(t) \cdot D^q h(t) \tag{3}$$

**Lemma 2.** [40]: *Suppose $h(t) \in R^n$ is a continuous and derivable function. Then, for $t \geq t_0$, we have:*

$$D^q h^T(t) h(t) \leq 2h^T(t) \cdot D^q h(t) \tag{4}$$

**Theorem 1.** [41]: *Suppose x = 0 is the equilibrium point of the fractional-order system (5) and its definition domain includes the origin. Suppose $V(t.x(t))$ is continuous and derivable and that the Lipschitz function is relative to x such that:*

$$D^q x(t) = f(x.t) \tag{5}$$

$$a_1 \|x\|^a \leq V(t.x(t)) \leq a_2 \|x\|^{ab} \tag{6}$$

$$D^q V(t.x(t)) \leq -a_3 \|x\|^{ab} \tag{7}$$

where $0 < q < 1$ and $a_1$, $a_2$, $a_3$, $a$, $b$ are arbitrary and positive constants. Then, $x = 0$ is stable in the sense of Mittag-Leffler.

**Definition 1.** *The continuous function $p : [0, \infty) \to [0, \infty)$ belongs to the class-K if it is strictly increasing and $p(0) = 0$.*

**Theorem 2.** [41]: *Suppose $x = 0$ is the equilibrium point of the fractional-order system (5), where $f(x.t)$ satisfies the Lipshitz condition and $q \in (0, 1)$. If the relations (8) and (9) are established for the Lyapunov function V(t,x(t)) and the class-K functions $\delta_i$:*

$$\delta_1(\|x\|) \leq V(t.x(t)) \leq \delta_2(\|x\|) \tag{8}$$

$$D^q V(t.x(t)) \leq -\delta_3(\|x\|) \tag{9}$$

Then, the system (5) is asymptotically stable in the sense of Mittag-Leffler [42].

**Theorem 3.** [43]: *For the fractional-order system (5) and the Lyapunov function $V(x)$:*

$$D^q V(x) \leq \left(\frac{\partial V}{\partial x}\right)^T \cdot D^q x = \left(\frac{\partial V}{\partial x}\right)^T \cdot f(x.t) \tag{10}$$

*2.2. Adaptive Synchronization between One Drive System and Several Response Systems*

Figure 1 shows the synchronization between one master system and several slave systems.

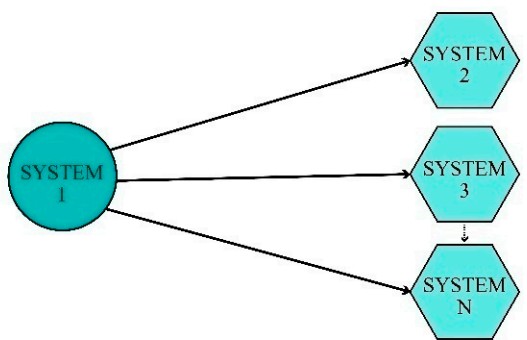

**Figure 1.** Transmission multi-mode synchronization.

The master hyper-chaotic system with unknown parameters is as follows [44]:

$$D^q x_1(t) = f_1(x_1) + H_1(x_1)\theta_1(t) \tag{11}$$

where $x_1(t) = [x_{11}, x_{12}, \cdots, x_{1n}]^T$ are the state vectors of the system, $f_1(x_1(t)) = [f_{11}, f_{12}, \cdots, f_{1n}]^T$ is a continuous function, $H_1(x_1(t)) = [H_{11}, H_{12}, \cdots, H_{1n}]^T$ is the matrix function, and $\theta_1(t) = [\theta_{11}, \theta_{12}. \cdots .\theta_{1n}]^T$ are the basic parameters of the master system that are unknown and variable.

N-1 slave hyper-chaotic systems with control function are as follows [44]:

$$D^q x_i(t) = f_i(x_i) + H_i(x_i)\theta_i(t) + u_{i-1}(t) \qquad i = 2, 3, \cdots, N \tag{12}$$

where $x_i(t) = [x_{i1}, x_{i2}, \cdots, x_{in}]^T$ is the state vector of $i^{\text{th}}$ system, $f_i(x_1(t)) = [f_{i1}, f_{i2}, \cdots, f_{in}]^T$ is the continuous function, $H_i(x_i(t)) = [H_{i1}, H_{i2}, \cdots, H_{in}]^T$ is the matrix function, $\theta_i(t) = [\theta_{i1}, \theta_{i2}, \cdots, \theta_{in}]^T$ are the basic parameters of $i^{\text{th}}$ slave system, and $u_{i-1}(t) = [u_{i-1.1}(t), u_{i-1.2}(t), \cdots, u_{i-1.n}(t)]^T$ is the control function of $i^{\text{th}}$ slave system. Therefore, according to Equations (11) and (12), the synchronization of the chaotic system with the control function is stated as follows:

$$\begin{cases} D^q x_1(t) = f_1(x_1) + H_1(x_1)\theta_1 \\ D^q x_2(t) = f_2(x_2) + H_2(x_2)\theta_2 + u_1(t) \\ \qquad\qquad \vdots \\ D^q x_N(t) = f_N(x_N) + H_N(x_N)\theta_N + u_{N-1}(t) \end{cases} \tag{13}$$

In multimode synchronization form, the synchronization error is given as follows:

$$e_{i-1}(t) = x_i(t) - x_1(t) \; i = 2, 3, \cdots, N$$

**Definition 2.** *For N FOCS expressed by (13), if adaptive controllers $u_{i-1}(t)$ exist such that for dynamic systems the error is given by:*

$$D^q e_{i-1}(t) = f_i(x_i) - f_1(x_1) + H_i(x_i)\theta_i - H_1(x_1)\theta_1 + u_{i-1}(t) \qquad i = 2, 3, \cdots, N-1 \tag{14}$$

The required conditions are:

$$\lim_{t \to \infty} \|e_{i-1}(t)\| = \lim_{t \to \infty} \|x_i(t) - x_1(t)\| \to 0 \; i = 2, 3, \cdots, N$$

If satisfied, then the adaptive transmission multi-mode synchronization between $N$ chaotic systems with unknown parameters is realized. The design of controllers and adaptive rules to achieve the above objective is based on the Lyapunov function, and synchronization is fulfilled with transmission mode synchronization. The controller law for $u_1(t), u_2(t), u_3(t), \cdots, u_{N-1}(t)$ is designed as below:

$$u_{i-1}(t) = -f_i(x_i) + f_1(x_1) - H_i(x_i)\hat{\theta}_i + H_1(x_1)\hat{\theta}_1 + K_{i-1}e_{i-1} \; i = 2, 3, \cdots, N-1 \tag{15}$$

Therefore, the errors of dynamics are given as follows:

$$D^q e_{i-1}(t) = H_i(x_i)\widetilde{\theta}_i - H_1(x_1)\widetilde{\theta}_1 + K_{i-1}e_{i-1} \; i = 2, 3, \cdots, N-1 \tag{16}$$

where $\hat{\theta}_i$ is the estimation of $\theta_i, \widetilde{\theta}_i(t) = \theta_i(t) - \hat{\theta}_i(t)$ is an approximation error, and:

$$K_{i-1} = diag(k_{i-1,1}.k_{i-1,2}.\cdots.k_{i-1,n}). \; k_{i-1,j} < 0 \; j = 1, 2, \cdots, n$$

*2.3. Adaptive Circular Multimode Synchronization of Chaotic Systems*

Figure 2 depicts a demonstration of circular multi-mode synchronizations. In this type of synchronization, all systems (except system 1) play both slave and master roles contemporarily, which helps to achieve more complexity.

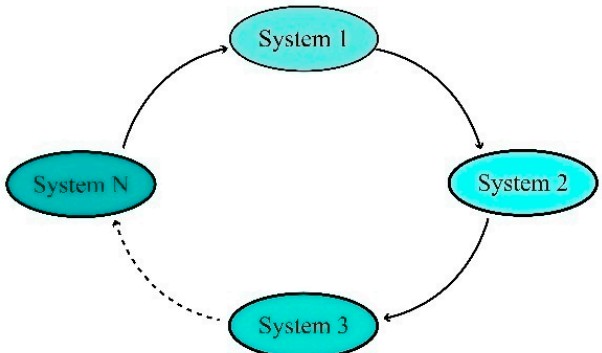

**Figure 2.** Circular multi-mode synchronizations.

In this case, the first chaotic system is described as the following.

$$D^q x_1(t) = f_1(x_1) + H_1(x_1)\theta_1 \tag{17}$$

The N-1 chaotic systems are given as follows [44]:

$$\begin{cases} D^q x_1(t) = f_1(x_1) + H_1(x_1)\theta_1 \\ D^q x_2(t) = f_2(x_2) + H_2(x_2)\theta_2 + m_1(t) \\ \qquad\qquad \vdots \\ D^q x_N(t) = f_N(x_N) + H_N(x_N)\theta_N + m_{N-1}(t) \end{cases} \tag{18}$$

Control input is in the form of $m_{i-1}(t) = [m_{i-1.1}.m_{i-1.2}.\cdots.m_{i-1.n}]^T$.

For $N$ chaotic system explained by (18), if there are adaptive controllers such that the error dynamic systems are given as follows:

$$r_i(t) = x_{i+1}(t) - x_i(t)i = 1,2,3,\cdots N-1$$

$$\begin{aligned} D^q r_{i-1}(t) &= f_i(x_i) - f_{i-1}(x_{i-1}) + H_i(x_i)\theta_i - H_{i-1}(x_{i-1})\theta_{i-1} + m_{i-1} - m_{i-2} \\ i &= 2,3,\cdots,N-1 \end{aligned} \tag{19}$$

In addition, conditions are given by:

$$\lim_{t\to\infty}\|r_{i-1}(t)\| = \lim_{t\to\infty}\|x_i(t) - x_{i-1}(t)\| \to 0 \qquad i = 2,3,\cdots,N$$

If satisfied, then an adaptive circular multi-mode synchronization between $N$ chaotic systems with unknown parameters is accomplished.

**Theorem 4.** *For multi-mode synchronization of hyper-chaotic in two circular and transitional states, the following statements are satisfied.*

**1-A**. If the transmission and circular synchronization are established with the controllers $u_i(t)$ and $m_i(t)$, then:

$$\forall i.j \ : \ \lim_{t\to\infty}\|x_i(t) - x_j(t)\| \to 0$$

**2-A**. If transmission synchronization is established, then circular synchronization is also realized and vice versa.

**Proof 1-A.** Suppose there is a transmission synchronization, therefore: $\forall\, i \ : \ e_i(t) \to 0$. Thus:$\square$

$$\forall i.j : \|x_i(t) - x_j(t)\| = \|(x_i(t) - x_1(t)) - (x_j(t) - x_1(t))\| \le \|e_i(t)\| + \|e_j(t)\| \to 0$$

If circular synchronization is met (assuming $i > j$):

$$\forall i.j : \|x_i(t) - x_j(t)\| = \|(x_i(t) - x_{i-1}(t)) + (x_{i-1}(t) - x_{i-2}(t)) + \cdots + (x_{j+1}(t) - x_j(t))\|$$
$$\leq \|r_{i-1}(t)\| + \cdots + \|r_j(t)\| = \sum_{k=j}^{i-1} \|r_k(t)\| \to 0 \tag{20}$$

**Proof 2-A.** The following relationships between errors exist in two synchronization modes:□

$$r_i(t) = \begin{cases} e_i(t) - e_{i-1}(t) & i = 2, 3, \cdots, N-1 \\ e_i(t) & i = 1 \end{cases} \quad . \, e_i(t) = \sum_{k=1}^{i} r_k(t)$$

If the transmission synchronization is true, $e_i(t) \to 0$; therefore:

$$\|r_i(t)\| = \begin{cases} \|e_i(t) - e_{i-1}(t)\| \leq \|e_i(t)\| + \|e_{i-1}(t)\| \to 0 & i \geq 2 \\ \|e_1(t)\| \to 0 & i = 1 \end{cases}$$

So, circular synchronization is true.
Conversely, assume that circular synchronization is true; $r_i(t) \to 0$. Therefore:

$$\|e_i(t)\| = \|\sum_{k=1}^{i} r_k(t)\| \leq \sum_{k=1}^{i} \|r_k(t)\| \to 0$$

Therefore, transmission synchronization is also true. So, both types of synchronization are equivalent to each other.

**Theorem 5.** *The control law is identical for transmission synchronization $u_i(t)$ and circular synchronization $m_i(t)$ of the fractional-order of chaotic systems.*

**Proof.** Utilizing the relationship between two types of errors, we have:□

$$D^q e_i(t) = D^q e_{i-1}(t) + D^q r_i(t)$$

Which can be acquired by substituting in Equations (14) and (23):

$$u_{i-1}(t) = u_i(t) - (m_i(t) - m_{i-1}(t)) \, i \geq 2 \tag{21}$$

For $i = 1$, we have:

$$e_1(t) = r_1(t) \Rightarrow D^q e_1(t) = D^q r_1(t) \Rightarrow m_1(t) = u_1(t) \tag{22}$$

By the substitution of (22) in (21) we have: $m_i(t) = u_i(t)$; hence, the proof is complete.

*2.4. Synchronization in the Presence of Disturbance, Time Delay and Uncertainty in the Systems*

In this case, the master and slave system with disturbance, time delay, and uncertainty is as follows:

$$\begin{cases} D^q x_1(t) = f_1(x_1) + H_1(x_1)\theta_1 + +F_1(x_1(t - \tau_1)) + \Delta f_1(x_1) + D_1(t) \\ D^q x_2(t) = f_2(x_2) + H_2(x_2)\theta_2 + F_2(x_2(t - \tau_2)) + \Delta f_2(x_2) + D_2(t) + u_1(t) \\ \quad\quad\quad\quad\quad\quad\quad \vdots \\ D^q x_N(t) = f_N(x_N) + H_N(x_N)\theta_N + F_N(x_N(t - \tau_N)) + \Delta f_N(x_N) + D_N(t) + u_{N-1}(t) \end{cases} \tag{23}$$

It is assumed that $F_i(x_i(t - \tau_i))$ are Lipschitz, uncertainties and disturbances are bounded but with an unknown boundary.

$$|\Delta f_i(x_i)| \le \gamma_i g_i(x_i). \qquad |D_i(t)| \le d_i \ i = 1, 2, \dots, N$$

$$|F_i(x_i(t - \tau_i)) - F_i(x_i(t - p_i))| \le l_i|\tau_i - p_i|$$

where $\gamma_i$, $l_i$ and $d_i$ are constant but unknown and $g_i(x_i)$ is definite and positive. The error dynamic is described as follows:

$$D^q e_{i-1}(t) = f_i(x_i) - f_1(x_1) + H_i(x_i)\theta_i - H_1(x_1)\theta_1 + F_i(x_i(t - \tau_i)) - F_1(x_1(t - \tau_1)) + \Delta f_i(x_i) - \Delta f_1(x_1) + D_i(t)$$
$$-D_1(t) + u_{i-1}(t) \qquad i = 2, 3, \cdots, N - 1 \tag{24}$$

Defining the control function as follows:

$$u_{i-1}(t) = -f_i(x_i) + f_1(x_1) - H_i(x_i)\hat{\theta}_i + H_1(x_1)\hat{\theta}_1 + K_{i-1}e_{i-1} - F_i(x_i(t - \hat{\tau}_i)) + F_1(x_1(t - \hat{\tau}_1)) + \bar{u}_{i-1}(t)i$$
$$= 2, 3, \cdots, N - 1 \tag{25}$$

where $\hat{\theta}_i$, $\hat{\tau}_i$ are estimations of $\theta_i$, $\tau_i$ and $\bar{u}_{i-1}(t)$ is the section of the control function, which is introduced below. By placing the control function in (24), the error dynamics are given as follows:

$$D^q e_{i-1}(t) = H_i(x_i)\widetilde{\theta}_i - H_1(x_1)\widetilde{\theta}_1 + \Delta f_i(x_i) - \Delta f_1(x_1) + D_i(t) - D_1(t) + K_{i-1}e_{i-1} + F_i(x_i(t - \tau_i)) - F_1(x_1(t - \tau_1))$$
$$-F_i(x_i(t - \hat{\tau}_i)) + F_1(x_1(t - \hat{\tau}_1)) + \bar{u}_{i-1}(t), \ i = 2, 3, \cdots, N - 1 \tag{26}$$

**Theorem 6.** *The error dynamics system (26) is under control law (36), the update rules (35) are stable, and the synchronization errors converge to zero despite the uncertainty, time delay, and disturbance.*

**Proof.** By explaining the Lyapunov function as follows:□

$$V = \frac{1}{2}(V_e + V_\theta + V_\gamma + V_d + V_\tau) \tag{27}$$

Wherein:

$$V_e = \sum_{i=2}^{N} e_{i-1}^T K_{i-1} e_{i-1} V_\theta = \sum_{i=2}^{N} \widetilde{\theta}_i^T \widetilde{\theta}_i + \widetilde{\theta}_1^T \widetilde{\theta}_1$$

$$V_\gamma = \sum_{i=2}^{N} \widetilde{\gamma}_i^2 + \widetilde{\gamma}_1^2, \ V_d = \sum_{i=2}^{N} \widetilde{d}_i^2 + \widetilde{d}_1^2, V_\tau = \sum_{i=2}^{N} l_i \widetilde{\tau}_i^2 + l_1 \widetilde{\tau}_1^2$$

and: $\widetilde{\theta}_i = \theta_i - \hat{\theta}_i$, $\widetilde{\gamma}_i = \gamma_i - \hat{\gamma}_i$, $\widetilde{d}_i = d_i - \hat{d}_i$, $\widetilde{\tau}_i = \tau_i - \hat{\tau}_i$.

By calculating the fractional derivative of the Lyapunov function and replacing control function (27):

$$D^q V \le \sum_{i=2}^{N} [e_{i-1}^T \Big( H_i(x_i)\widetilde{\theta}_i - H_1(x_1)\widetilde{\theta}_1 + \Delta f_i(x_i) - \Delta f_1(x_1) + F_i(x_i(t - \tau_i)) - F_i(x_i(t - \hat{\tau}_i)) - F_1(x_1(t - \tau_1))$$
$$+F_1(x_1(t - \hat{\tau}_1)) + D_i(t) - D_1(t) + \tfrac{1}{2}(K_{i-1} + K_{i-1}^T)e_{i-1} \Big) + \widetilde{\theta}_i^T D^q \widetilde{\theta}_i + \widetilde{\gamma}_i D^q \widetilde{\gamma}_i + \widetilde{d}_i D^q \widetilde{d}_i + l_i \widetilde{\tau}_i D^q \tau_i \tag{28}$$
$$+\bar{u}_{i-1}(t)] + \widetilde{\theta}_1^T D^q \widetilde{\theta}_1 + \widetilde{\gamma}_1 D^q \widetilde{\gamma}_1 + \widetilde{d}_1 D^q \widetilde{d}_1 + l_1 \widetilde{\tau}_1 D^q \tau_1$$

If $\bar{u}_i{}^j$, $\Delta f_i{}^j$, $D_i{}^j$, $e_{i-1}^j$ is the component *j*th of the vectors $\bar{u}_{i-1}(t)$, $\Delta f_i$, $D_i$, $e_{i-1}$, respectively. Then:

$$D^q V \leq \sum_{i=2}^{N} \sum_{j=1}^{n} e_{i-1}^j (\Delta f_i{}^j - \Delta f_1{}^j + D_i{}^j - D_1{}^j + F_i(x_i(t-\tau_i)) - F_i(x_i(t-\hat{\tau}_i)) - F_1(x_1(t-\tau_1)) + F_1(x_1(t-\hat{\tau}_1))$$

$$+ \bar{u}_{i-1}{}^j) + \sum_{i=2}^{N} [e_{i-1}^T \left( H_i(x_i)\widetilde{\theta}_i - H_1(x_1)\widetilde{\theta}_1 \right) + \widetilde{\theta}_i^T D^q \widetilde{\theta}_i] + \sum_{i=2}^{N} \left( \widetilde{\gamma}_i D^q \widetilde{\gamma}_i + \widetilde{d}_i D^q \widetilde{d}_i + l_i \widetilde{\tau}_i D^q \tau_i \right) \tag{29}$$

$$+ \sum_{i=2}^{N} \tfrac{1}{2} e_{i-1}^T \left( K_{i-1} + K_{i-1}{}^T \right) e_{i-1} + \widetilde{\gamma}_1 D^q \widetilde{\gamma}_1 + \widetilde{d}_1 D^q \widetilde{d}_1 + l_1 \widetilde{\tau}_1 D^q \tau_1 + \widetilde{\theta}_1^T D^q \widetilde{\theta}_1$$

Therefore:

$$D^q V \leq \sum_{i=2}^{N} \sum_{j=1}^{n} \left[ \left| e_{i-1}^j \right| \left( \left| \Delta f_i{}^j \right| + \left| \Delta f_1{}^j \right| + \left| D_i{}^j \right| + \left| D_1{}^j \right| + |F_i(x_i(t-\tau_i)) - F_i(x_i(t-\hat{\tau}_i))| + |-F_1(x_1(t-\tau_1)) + F_1(x_1(t-\hat{\tau}_1))| \right) \right.$$

$$+ e_{i-1}^j \bar{u}_{i-1}{}^j] + \sum_{i=2}^{N} \left( \widetilde{\gamma}_i D^q \widetilde{\gamma}_i + \widetilde{d}_i D^q \widetilde{d}_i + l_i \widetilde{\tau}_i D^q \tau_i \right) + \sum_{i=2}^{N} e_{i-1}^T K_{i-1} e_{i-1} + \sum_{i=2}^{N} [e_{i-1}^T \left( H_i(x_i)\widetilde{\theta}_i - H_1(x_1)\widetilde{\theta}_1 \right) \tag{30}$$

$$+ \widetilde{\theta}_i^T D^q \widetilde{\theta}_i] + \widetilde{\gamma}_1 D^q \widetilde{\gamma}_1 + \widetilde{d}_1 D^q \widetilde{d}_1 + l_1 \widetilde{\tau}_1 D^q \tau_1 + \widetilde{\theta}_1^T D^q \widetilde{\theta}_1$$

The disturbance and uncertainty boundary condition can be extended to the components of $\Delta f_i$ and $D_i(t)$ as follows:

$$\left| \Delta f_i{}^j \right| \leq \max_j \left| \Delta f_i{}^j \right| \leq |\Delta f_i(x_i)| \leq \gamma_i g_i(x_i)$$

$$\left| D_i{}^j(t) \right| \leq \max_j \left| D_i{}^j(t) \right| \leq |D_i(t)| \leq d_i$$

which we have by substituting in (30):

$$D^q V \leq \sum_{i=2}^{N} \sum_{j=1}^{n} \left[ \left| e_{i-1}^j \right| (\gamma_i g_i(x_i) + \gamma_1 g_1(x_1) + d_i + d_1 + l_i|\widetilde{\tau}_i|) + e_{i-1}^j \bar{u}_{i-1}{}^j \right] + \sum_{i=2}^{N} \left( \widetilde{\gamma}_i D^q \widetilde{\gamma}_i + \widetilde{d}_i D^q \widetilde{d}_i + l_i \widetilde{\tau}_i D^q \tau_i \right)$$

$$+ \sum_{i=2}^{N} e_{i-1}^T K_{i-1} e_{i-1} + \sum_{i=2}^{N} [e_{i-1}^T \left( H_i(x_i)\widetilde{\theta}_i - H_1(x_1)\widetilde{\theta}_1 \right) + \widetilde{\theta}_i^T D^q \widetilde{\theta}_i] + \widetilde{\gamma}_1 D^q \widetilde{\gamma}_1 + \widetilde{d}_1 D^q \widetilde{d}_1 + l_1 \widetilde{\tau}_1 D^q \tau_1 \tag{31}$$

$$+ \widetilde{\theta}_1^T D^q \widetilde{\theta}_1$$

If $\bar{u}_{i-1}{}^j(t)$ is defined as follows:

$$\bar{u}_{i-1}{}^j(t) = -(\hat{\gamma}_i g_i(x_i) + \hat{\gamma}_1 g_1(x_1) + \hat{d}_i + \hat{d}_1 + \hat{l}_i + \hat{l}_1) \cdot sgn\left( e_{i-1}^j(t) \right) \tag{32}$$

Through the estimation of disturbance and uncertainty bounds in $\bar{u}_{i-1}{}^j(t)$, an effort was made to eliminate the effects of disturbance, delay, and uncertainty as much as possible. Therefore, Lyapunov function derivatives will be negative by selecting the proper update rules, and, ultimately, the convergence of the synchronization error to zero will be guaranteed.

Then,

$$D^q V \leq \sum_{i=2}^{N} \sum_{j=1}^{n} \left[ \left| e_{i-1}^j \right| \left( \widetilde{\gamma}_i g_i(x_i) + \widetilde{\gamma}_1 g_1(x_1) + \widetilde{d}_i + \widetilde{d}_1 + l_i|\widetilde{\tau}_i| \right) \right] + \sum_{i=2}^{N} \left( \widetilde{\gamma}_i D^q \widetilde{\gamma}_i + \widetilde{d}_i D^q \widetilde{d}_i + l_i \widetilde{\tau}_i D^q \tau_i \right) + \sum_{i=2}^{N} e_{i-1}^T K_{i-1} e_{i-1}$$

$$+ \sum_{i=2}^{N} [e_{i-1}^T \left( H_i(x_i)\widetilde{\theta}_i - H_1(x_1)\widetilde{\theta}_1 \right) + \widetilde{\theta}_i^T D^q \widetilde{\theta}_i] + \widetilde{\gamma}_1 D^q \widetilde{\gamma}_1 + \widetilde{d}_1 D^q \widetilde{d}_1 + l_1 \widetilde{\tau}_1 D^q \tau_1 + \widetilde{\theta}_1^T D^q \widetilde{\theta}_1 \tag{33}$$

The rules updating estimation errors are as follows:

$$D^q \widetilde{\gamma}_i = -(g_i(x_i) \sum_{j=1}^{n} \left| e_{i-1}^j \right| + \alpha_i \widetilde{\gamma}_i), \qquad i = 2, 3, \ldots, N \tag{34}$$

$$D^q \widetilde{\gamma}_1 = -\left( \sum_{i=1}^{N} \sum_{j=1}^{n} \left| e_{i-1}^j \right| g_1(x_1) + \alpha_1 \widetilde{\gamma}_1 \right) \tag{35}$$

$$D^q \widetilde{d}_i = -\left( \sum_{j=1}^{n} \left| e_{i-1}^j \right| + \beta_i \widetilde{d}_i \right), \qquad i = 2, 3, \ldots, N \tag{36}$$

$$D^q \tilde{d}_1 = -\left( \sum_{i=1}^{N} \sum_{j=1}^{n} \left| e_{i-1}^j \right| + \beta_1 \tilde{d}_1 \right) \tag{37}$$

$$D^q \tilde{\tau}_i = -\left( \sum_{j=1}^{n} \left| e_{i-1}^j \right| + \mu_i \tilde{\tau}_i \right), \qquad i = 2, 3, \dots, N \tag{38}$$

$$D^q \tilde{\tau}_1 = -\left( \sum_{i=1}^{N} \sum_{j=1}^{n} \left| e_{i-1}^j \right| + \mu_1 \tilde{\tau}_1 \right) \tag{39}$$

$$D^q \tilde{\theta}_i = -H_i^T(x_i) e_{i-1} - \sigma_i \tilde{\theta}_i, \; i = 2, 3, \dots, N \tag{40}$$

$$D^q \tilde{\theta}_1 = -\sum_{i=2}^{N} H_1^T(x_1) e_{i-1} - \sigma_1 \tilde{\theta}_1 \tag{41}$$

where $\alpha_i$, $\beta_i$, $\sigma_i$, $\mu_i$ are positive. By placing the above update rules in (33), we have:

$$D^q V \leq \sum_{i=2}^{N} e_{i-1}^T K_{i-1} e_{i-1} - \sum_{i=1}^{N} (\alpha_i \tilde{\gamma}_i^2 + \beta_i \tilde{d}_i^2 + \mu_i \tilde{\tau}_i^2) - \sum_{i=1}^{N} \sigma_i \tilde{\theta}_i^T \tilde{\theta}_i < -\mu V$$

where: $\mu = \min_{i,j} \left( \alpha_i, \beta_i, \sigma_i, \mu_i, -k_{i-1,j} \right) > 0$. Therefore, according to the Theorems 1 and 2 and being Hurwitz $K_{i-1}$, the stability of the system according to Mittag-Leffler is also confirmed. The convergence of synchronization errors to zero is also guaranteed despite uncertainty and disturbance. The estimations updating rules are obtained as follows:

$$D^q \hat{\gamma}_i = g_i(x_i) \sum_{j=1}^{n} \left| e_{i-1}^j \right| + \alpha_i \tilde{\gamma}_i, \qquad i = 2, 3, \dots, N \tag{42}$$

$$D^q \hat{\gamma}_1 = g_1(x_1) \sum_{i=1}^{N} \sum_{j=1}^{n} \left| e_{i-1}^j \right| + \alpha_1 \tilde{\gamma}_1. \tag{43}$$

$$D^q \hat{d}_i = \sum_{j=1}^{n} \left| e_{i-1}^j \right| + \beta_i \tilde{d}_i, \quad i = 2, 3, \dots, N. \tag{44}$$

$$D^q \hat{d}_1 = \sum_{i=1}^{N} \sum_{j=1}^{n} \left| e_{i-1}^j \right| + \beta_1 \tilde{d}_1. \tag{45}$$

$$D^q \hat{\tau}_i = \sum_{j=1}^{n} \left| e_{i-1}^j \right| + \mu_i \tilde{\tau}_i, \qquad i = 2, 3, \dots, N \tag{46}$$

$$D^q \hat{\tau}_1 = \sum_{i=1}^{N} \sum_{j=1}^{n} \left| e_{i-1}^j \right| + \mu_1 \tilde{\tau}_1 \tag{47}$$

Therefore, the final control function is as follows:

$$u_{i-1}(t) = -f_i(x_i) + f_1(x_1) - H_i(x_i)\hat{\theta}_i(t) + H_1(x_1)\hat{\theta}_1(t) + K_{i-1}e_{i-1}(t) - F_i(x_i(t-\hat{\tau}_i)) + F_1(x_1(t-\hat{\tau}_1)) - (\hat{\gamma}_i g_i(x_i) + \hat{\gamma}_1 g_1(x_1) + \hat{d}_i \\ + \hat{d}_1 + \hat{l}_i + \hat{l}_1) \cdot sgn\left(e_{i-1}^j(t)\right), \qquad i = 1, 2, \cdots, N-1 \tag{48}$$

Selecting $\bar{u}_{i-1}{}^j(t)$ as (36) due to the presence of the discontinuous function $sgn\left(e_{i-1}^j(t)\right)$ causes discontinuity in the control function (48). This problem will be solved by correcting Equation (32) and update rules (42–47), as well as Theorem 7.

**Theorem 7.** *If $\overline{u}_{i-1}{}^j(t)$ is selected as $\overline{u}_{i-1}{}^j(t) = -(\hat{\gamma}_i g_i(x_i) + \hat{\gamma}_1 g_1(x_1) + \hat{d}_i + \hat{d}_1 + \hat{l}_i + \hat{l}_1) \cdot sat\left(e_{i-1}^j, \lambda\right)$, then the control function $u_{i-1}(t)$ will become continuous and the convergence of synchronization errors would be guaranteed to be at zero. Further, update rules will be as follows:*

$$D^q \widetilde{\gamma}_i = -g_i(x_i) - \alpha_i \widetilde{\gamma}_i - \frac{\hat{\gamma}_i g_i(x_i)}{\hat{\gamma}_i} \varphi_{i-1}(t).$$

$$D^q \widetilde{d}_i = -\sum_{j=1}^{n} \left| e_{i-1}^j \right| - \beta_i \widetilde{d}_i - \frac{\hat{d}_i}{\hat{d}_i} \varphi_{i-1}(t)$$

$$D^q \widetilde{\tau}_i = -\sum_{j=1}^{n} \left| e_{i-1}^j \right| - \mu_i \widetilde{\tau}_i - \frac{\hat{\tau}_i}{\hat{\tau}_i} \varphi_{i-1}(t)$$

$$D^q \widetilde{\gamma}_1 = -g_1(x_1) \sum_{i=2}^{N} \sum_{j=1}^{n} \left| e_{i-1}^j \right| - \alpha_1 \widetilde{\gamma}_1 - \frac{\hat{\gamma}_1 g_1(x_1)}{\hat{\gamma}_1} \sum_{i=2}^{N} \varphi_{i-1}(t)$$

$$D^q \widetilde{d}_1 = -\sum_{i=2}^{N} \sum_{j=1}^{n} \left| e_{i-1}^j \right| - \beta_1 \widetilde{d}_1 - \frac{\hat{d}_1}{\hat{d}_1} \sum_{i=2}^{N} \varphi_{i-1}(t) \tag{49}$$

$$D^q \widetilde{\tau}_1 = -\sum_{i=2}^{N} \sum_{j=1}^{n} \left| e_{i-1}^j \right| - \mu_1 \widetilde{\tau}_1 - \frac{\hat{\tau}_1}{\hat{\tau}_1} \sum_{i=2}^{N} \varphi_{i-1}(t)$$

$$D^q \hat{\gamma}_i = -D^q \widetilde{\gamma}_i, \ \ D^q \hat{d}_i = -D^q \widetilde{d}_i, \ \ D^q \hat{\tau}_i = -D^q \widetilde{\tau}_i$$

*where $\lambda$ and $\varepsilon$ are small positive numbers and:*

$$sat\left(e_{i-1}^j, \lambda\right) = \begin{cases} sgn\left(e_{i-1}^j(t)\right), & \left| e_{i-1}^j(t) \right| \geq \lambda \\ \frac{e_{i-1}^j(t)}{\lambda}, & \left| e_{i-1}^j(t) \right| \leq \lambda \end{cases}$$

$$\varphi_{i-1}(t) = \sum_{j=1}^{n} \left[ \left| e_{i-1}^j \right| - e_{i-1}^j \cdot sat\left(e_{i-1}^j, \lambda\right) \right]$$

**Proof.** Through Equation (31) and by placing $\overline{u}_{i-1}{}^j(t) = -(\hat{\gamma}_i g_i(x_i) + \hat{\gamma}_1 g_1(x_1) + \hat{d}_i + \hat{d}_1 + \hat{l}_i + \hat{l}_1) \cdot sat\left(e_{i-1}^j, \lambda\right)$ and $\gamma_i = \widetilde{\gamma}_i + \hat{\gamma}_i, \tau_i = \widetilde{\tau}_i + \hat{\tau}_i, \ d_i = \widetilde{d}_i + \hat{d}_i$, we will get:□

$$D^q V \leq \sum_{i=2}^{N} \sum_{j=1}^{n} \left[ \left| e_{i-1}^j \right| \left( (\widetilde{\gamma}_i + \hat{\gamma}_i) g_i(x_i) + (\widetilde{\gamma}_1 + \hat{\gamma}_1) g_1(x_1) + \left( \widetilde{d}_i + \hat{d}_i \right) + \left( \widetilde{d}_1 + \hat{d}_1 \right) + \left( \widetilde{l}_i + \hat{l}_i \right) + \left( \widetilde{l}_1 + \hat{l}_1 \right) \right) - e_{i-1}^j (\hat{\gamma}_i g_i(x_i) \right.$$

$$\left. + \hat{\gamma}_1 g_1(x_1) + \hat{d}_i + \hat{d}_1 + \hat{l}_i + \hat{l}_1 ) \cdot sat\left(e_{i-1}^j, \lambda\right) \right] + \sum_{i=2}^{N} \left( \widetilde{\gamma}_i D^q \widetilde{\gamma}_i + \widetilde{d}_i D^q \widetilde{d}_i + \widetilde{\tau}_i D^q \tau_i \right) + \sum_{i=2}^{N} e_{i-1}^T K_{i-1} e_{i-1} \tag{50}$$

$$+ \widetilde{\gamma}_1 D^q \widetilde{\gamma}_1 + \widetilde{d}_1 D^q \widetilde{d}_1 + \widetilde{\tau}_1 D^q \tau_1 - \sum_{i=1}^{N} \sigma_i \widetilde{\theta}_i^T \widetilde{\theta}_i$$

By classifying the function above as below:

$$D^q V \leq \sum_{i=2}^{N} \left( \sum_{j=1}^{n} \left[ \left| e_{i-1}^j \right| ((\widetilde{\gamma}_i + \hat{\gamma}_i) g_i(x_i)) - e_{i-1}^j \hat{\gamma}_i g_i(x_i) \cdot sat\left(e_{i-1}^j, \lambda\right) \right] + \widetilde{\gamma}_i D^q \widetilde{\gamma}_i \right)$$

$$+ \sum_{i=2}^{N} \left( \sum_{j=1}^{n} \left[ \left| e_{i-1}^j \right| \left( \widetilde{d}_i + \hat{d}_i \right) - e_{i-1}^j \hat{d}_i \cdot sat\left(e_{i-1}^j, \lambda\right) \right] + \widetilde{d}_i D^q \widetilde{d}_i \right)$$

$$+ \sum_{i=2}^{N} \left( \sum_{j=1}^{n} \left[ \left| e_{i-1}^j \right| (\widetilde{\tau}_i + \hat{\tau}_i) - e_{i-1}^j \hat{\tau}_i \cdot sat\left(e_{i-1}^j, \lambda\right) \right] + \widetilde{\tau}_i D^q \widetilde{\tau}_i \right)$$

$$+ \sum_{i=2}^{N} \sum_{j=1}^{n} \left[ \left| e_{i-1}^j \right| ((\widetilde{\gamma}_1 + \hat{\gamma}_1) g_1(x_1)) - e_{i-1}^j \hat{\gamma}_1 g_1(x_1) \cdot sat\left(e_{i-1}^j, \lambda\right) \right] \tag{51}$$

$$+ \widetilde{\gamma}_1 D^q \widetilde{\gamma}_1 + \sum_{i=2}^{N} \sum_{j=1}^{n} \left[ \left| e_{i-1}^j \right| \left( \widetilde{d}_1 + \hat{d}_1 \right) - e_{i-1}^j \hat{d}_1 \cdot sat\left(e_{i-1}^j, \lambda\right) \right] + \widetilde{d}_1 D^q \widetilde{d}_1$$

$$+ \sum_{i=2}^{N} \sum_{j=1}^{n} \left[ \left| e_{i-1}^j \right| (\widetilde{\tau}_1 + \hat{\tau}_1) - e_{i-1}^j \hat{\tau}_1 \cdot sat\left(e_{i-1}^j, \lambda\right) \right] + \widetilde{\tau}_1 D^q \widetilde{\tau}_1$$

$$+ \sum_{i=2}^{N} e_{i-1}^T K_{i-1} e_{i-1} - \sum_{i=1}^{N} \sigma_i \widetilde{\theta}_i^T \widetilde{\theta}_i$$

To determine update rules, we can act as below:

$$\sum_{j=1}^{n} \left[ \left| e_{i-1}^j \right| ((\widetilde{\gamma}_i + \hat{\gamma}_i) g_i(x_i)) - e_{i-1}^j \hat{\gamma}_i g_i(x_i) \cdot sat\left(e_{i-1}^j, \lambda\right) \right] + \widetilde{\gamma}_i D^q \widetilde{\gamma}_i = -\alpha_i \widetilde{\gamma}_i{}^2$$

$$\sum_{j=1}^{n} \left[ \left| e_{i-1}^j \right| \left( \widetilde{d}_i + \hat{d}_i \right) - e_{i-1}^j \hat{d}_i \cdot sat\left(e_{i-1}^j, \lambda\right) \right] + \widetilde{d}_i D^q \widetilde{d}_i = -\beta_i \widetilde{d}_i{}^2$$

$$\sum_{i=2}^{N} \sum_{j=1}^{n} \left[ \left| e_{i-1}^j \right| ((\widetilde{\gamma}_1 + \hat{\gamma}_1) g_1(x_1)) - e_{i-1}^j \hat{\gamma}_1 g_1(x_1) \cdot sat\left(e_{i-1}^j, \lambda\right) \right] + \widetilde{\gamma}_1 D^q \widetilde{\gamma}_1 = -\alpha_1 \widetilde{\gamma}_1{}^2 \tag{52}$$

$$\sum_{i=2}^{N} \sum_{j=1}^{n} \left[ \left| e_{i-1}^j \right| \left( \widetilde{d}_1 + \hat{d}_1 \right) - e_{i-1}^j \hat{d}_1 \cdot sat\left(e_{i-1}^j, \lambda\right) \right] + \widetilde{d}_1 D^q \widetilde{d}_1 = -\beta_1 \widetilde{d}_1{}^2$$

$$\sum_{j=1}^{n} \left[ \left| e_{i-1}^j \right| (\widetilde{\tau}_i + \hat{\tau}_i) - e_{i-1}^j \hat{\tau}_i \cdot sat\left(e_{i-1}^j, \lambda\right) \right] + \widetilde{\tau}_i D^q \widetilde{\tau}_i = -\mu_i \widetilde{\tau}_i{}^2$$

$$\sum_{i=2}^{N}\sum_{j=1}^{n}\left[\left|e_{i-1}^{j}\right|(\tilde{\tau}_1+\hat{\tau}_1)-e_{i-1}^{j}\hat{\tau}_1\cdot sat\left(e_{i-1}^{j},\lambda\right)\right]+\tilde{\tau}_1 D^q\tilde{\tau}_1=-\mu_1\tilde{\tau}_1^{2}$$

Given the definition of function $\varphi_{i-1}(t)$, the obtained update rules will be as follows:

$$D^q\tilde{\gamma}_i=-g_i(x_i)-\alpha_i\tilde{\gamma}_i-\frac{\hat{\gamma}_i g_i(x_i)}{\tilde{\gamma}_i}]\varphi_{i-1}(t)$$
$$D^q\tilde{d}_i=-\sum_{j=1}^{n}\left|e_{i-1}^{j}\right|-\beta_i\tilde{d}_i-\frac{\hat{d}_i}{\tilde{d}_i}]\varphi_{i-1}(t)$$
$$D^q\tilde{\gamma}_1=-g_1(x_1)\sum_{i=2}^{N}\sum_{j=1}^{n}\left|e_{i-1}^{j}\right|-\alpha_1\tilde{\gamma}_1-\frac{\hat{\gamma}_1 g_1(x_1)}{\tilde{\gamma}_1}\sum_{i=2}^{N}\varphi_{i-1}(t)$$

$$D^q\tilde{d}_1=-\sum_{i=2}^{N}\sum_{j=1}^{n}\left|e_{i-1}^{j}\right|-\beta_1\tilde{d}_1-\frac{\hat{d}_1}{\tilde{d}_1}\sum_{i=2}^{N}\varphi_{i-1}(t)$$

$$D^q\tilde{\tau}_i=-\sum_{j=1}^{n}\left|e_{i-1}^{j}\right|-\mu_i\tilde{\tau}_i-\frac{\hat{\tau}_i}{\tilde{\tau}_i}\varphi_{i-1}(t)$$

$$D^q\tilde{\tau}_1=-\sum_{i=2}^{N}\sum_{j=1}^{n}\left|e_{i-1}^{j}\right|-\beta_1\tilde{\tau}_1-\frac{\hat{\tau}_1}{\tilde{\tau}_1}\sum_{i=2}^{N}\varphi_{i-1}(t)$$

(53)

In Equation (52), it is possible that each of the estimation errors $(\tilde{d}_i,\ \tilde{\gamma}_i,\tilde{\tau}_i,\ i=1,2,3,\cdots N)$ equal zero, in which case the update rules cannot be used. Therefore, to prevent denominators from tending to zero in the update rules, the rules will be corrected as follows:

$$D^q\tilde{\gamma}_i=-g_i(x_i)\sum_{j=1}^{n}\left|e_{i-1}^{j}\right|-\alpha_i\tilde{\gamma}_i-\frac{\tilde{\gamma}_i\hat{\gamma}_i g_i(x_i)}{\tilde{\gamma}_i^{2}+\varepsilon}\varphi_{i-1}(t)$$
$$D^q\tilde{d}_i=-\sum_{j=1}^{n}\left|e_{i-1}^{j}\right|-\beta_i\tilde{d}_i-\frac{\tilde{d}_i\hat{d}_i}{\tilde{d}_i^{2}+\varepsilon}\varphi_{i-1}(t)$$
$$D^q\tilde{\gamma}_1=-g_1(x_1)\sum_{i=2}^{N}\sum_{j=1}^{n}\left|e_{i-1}^{j}\right|-\alpha_1\tilde{\gamma}_1-\frac{\tilde{\gamma}_1\hat{\gamma}_1 g_1(x_1)}{\tilde{\gamma}_1^{2}+\varepsilon}\sum_{i=2}^{N}\varphi_{i-1}(t)$$
$$D^q\tilde{d}_1=-\sum_{i=2}^{N}\sum_{j=1}^{n}\left|e_{i-1}^{j}\right|-\beta_1\tilde{d}_1-\frac{\tilde{d}_1\hat{d}_1}{\tilde{d}_1^{2}+\varepsilon}\sum_{i=2}^{N}]\varphi_{i-1}(t)$$
$$D^q\tilde{\tau}_i=-\sum_{j=1}^{n}\left|e_{i-1}^{j}\right|-\mu_i\tilde{\tau}_i-\frac{\tilde{\tau}_i\hat{\tau}_i}{\tilde{\tau}_i^{2}+\varepsilon}\varphi_{i-1}(t)$$

$$D^q\tilde{\tau}_1=-\sum_{i=2}^{N}\sum_{j=1}^{n}\left|e_{i-1}^{j}\right|-\beta_1\tilde{\tau}_1-\frac{\tilde{\tau}_1\hat{\tau}_1}{\tilde{\tau}_1^{2}+\varepsilon}\sum_{i=2}^{N}\varphi_{i-1}(t)$$

(54)

where $\varepsilon$ is positive and very small in the equation above $(0<\varepsilon\ll 1)$.

In the rules above, if $\left|e_{i-1}^{j}\right|\geq\lambda\Rightarrow\varphi_{i-1}(t)=0$, they change to the previous rules (54). Hence, Equation (33) stands for $\left|e_{i-1}^{j}\right|\geq\lambda$.

Given the definition of equation $\varphi_{i-1}(t)$, we get:

$$|\varphi_{i-1}(t)|=\left\{\begin{array}{ll}0 & \left|e_{i-1}^{j}\right|\geq\lambda\\[4pt]\leq\frac{n}{4}\lambda & \left|e_{i-1}^{j}\right|<\lambda\end{array}\right.$$

(55)

By placing corrected update rules (52) in (39), we get:

$$D^q V\leq\sum_{i=2}^{N}e_{i-1}^{T}K_{i-1}e_{i-1}-\sum_{i=1}^{N}(\alpha_i\tilde{\gamma}_i^{2}+\beta_i\tilde{d}_i^{2}+\mu_i\tilde{\tau}_i^{2})-\sum_{i=1}^{N}\sigma_i\tilde{\theta}_i^{T}\tilde{\theta}_i+\rho(t)$$

(56)

where:

$$\rho(t)=\sum_{i=2}^{N}(\frac{\hat{\gamma}_i g_i(x_i)}{\tilde{\gamma}_i^{2}+\varepsilon}+\frac{\hat{d}_i}{\tilde{d}_i^{2}+\varepsilon}+\frac{\hat{\tau}_i}{\tilde{\tau}_i^{2}+\varepsilon})\varphi_{i-1}(t)+\varepsilon\left(\frac{\hat{\gamma}_1 g_1(x_1)}{\tilde{\gamma}_1^{2}+\varepsilon}+\frac{\hat{d}_i}{\tilde{d}_i^{2}+\varepsilon}+\frac{\hat{\tau}_i}{\tilde{\tau}_i^{2}+\varepsilon}\right)\sum_{i=2}^{N}\varphi_{i-1}(t)$$

Given condition $\left|e_{i-1}^{j}\right|\geq\lambda\Rightarrow\varphi_{i-1}(t)=0$, we get:

$$D^q V\leq\sum_{i=2}^{N}e_{i-1}^{T}K_{i-1}^{2}e_{i-1}-\sum_{i=1}^{N}(\alpha_i\tilde{\gamma}_i^{2}+\beta_i\tilde{d}_i^{2}+\mu_i\tilde{\tau}_i^{2})-\sum_{i=1}^{N}\sigma_i\tilde{\theta}_i^{T}\tilde{\theta}_i<-\mu V$$

Which indicates the norm reduction of $e_{i-1},\ \tilde{\theta}_i,\ \tilde{\gamma}_i,\ \tilde{d}_i$, and $\tilde{\tau}_i$. Therefore, the norm of synchronization errors is bounded.

In $\left|e_{i-1}^{j}\right|<\lambda$, we have:

$$|\rho(t)|=\left|\sum_{i=2}^{N}(\frac{\hat{\gamma}_i g_i(x_i)}{\tilde{\gamma}_i^{2}+\varepsilon}+\frac{\hat{d}_i}{\tilde{d}_i^{2}+\varepsilon}+\frac{\hat{\tau}_i}{\tilde{\tau}_i^{2}+\varepsilon})\varepsilon\varphi_{i-1}(t)+\varepsilon\left(\frac{\hat{\gamma}_1 g_1(x_1)}{\tilde{\gamma}_1^{2}+\varepsilon}+\frac{\hat{d}_i}{\tilde{d}_i^{2}+\varepsilon}+\frac{\hat{\tau}_i}{\tilde{\tau}_i^{2}+\varepsilon}\right)\sum_{i=2}^{N}\varphi_{i-1}(t)\right|$$
$$\leq\left|\sum_{i=2}^{N}(\frac{\hat{\gamma}_i g_i(x_i)}{\tilde{\gamma}_i^{2}+\varepsilon}+\frac{\hat{d}_i}{\tilde{d}_i^{2}+\varepsilon}+\frac{\hat{\tau}_i}{\tilde{\tau}_i^{2}+\varepsilon})\right|\frac{n}{4}\lambda\varepsilon+\left|\frac{\hat{\gamma}_1 g_1(x_1)}{\tilde{\gamma}_1^{2}+\varepsilon}+\frac{\hat{d}_i}{\tilde{d}_i^{2}+\varepsilon}+\frac{\hat{\tau}_i}{\tilde{\tau}_i^{2}+\varepsilon}\right|\frac{n(N-1)}{4}\lambda\varepsilon=\rho_0(t)\lambda\varepsilon$$

where: $\rho_0(t) = \left| \sum_{i=2}^{N} \left( \frac{\hat{\gamma}_i g_i(x_i)}{\hat{\gamma}_i^2 + \varepsilon} + \frac{\hat{d}_i}{\hat{d}_i^2 + \varepsilon} + \frac{\hat{\tau}_i}{\hat{\tau}_i^2 + \varepsilon} \right) \right| \frac{n}{4} + \left| \frac{\hat{\gamma}_1 g_1(x_1)}{\hat{\gamma}_1^2 + \varepsilon} + \frac{\hat{d}_i}{\hat{d}_i^2 + \varepsilon} + \frac{\hat{\tau}_i}{\hat{\tau}_i^2 + \varepsilon} \right| \frac{n(N-1)}{4}$ and bounded. Therefore:

$$D^q V \leq \sum_{i=2}^{N} e_{i-1}^T K_{i-1} e_{i-1} - \sum_{i=1}^{N} (\alpha_i \widetilde{\gamma}_i^2 + \beta_i \widetilde{\gamma}_i^2 + \mu_i \widetilde{\tau}_i^2) - \sum_{i=1}^{N} \sigma_i \widetilde{\theta}_i^T \widetilde{\theta}_i + \rho_0(t) \lambda \varepsilon$$

If the tracking error enters zone $\left| e_{i-1}^j \right| < \lambda$, it will remain there because, as soon as it reaches that zone, $\left| e_{i-1}^j \right| = \lambda$. Given condition $\varphi_{i-1}(t) = 0 \Rightarrow \rho(t) = 0$ and Equation (54), the Lyapunov function derivative in the equation is as follows:

$$D^q V < -\mu V$$

Which indicates a decrease in all errors. Hence, by selecting $\lambda$ and $\varepsilon$, which are sufficiently small, synchronization errors would consistently be very small, and synchronization will be conducted with proper precision. Therefore, the final control function is as follows:

$$\begin{aligned} u_{i-1}(t) = &-f_i(x_i) + f_1(x_1) - H_i(x_i)\hat{\theta}_i(t) + H_1(x_1)\hat{\theta}_1(t) - F_i(x_i(t - \hat{\tau}_i)) + F_1(x_1(t - \hat{\tau}_1)) + K_{i-1}e_{i-1}(t) \\ &- (\hat{\gamma}_i g_i(x_i) + \hat{\gamma}_1 g_1(x_1) + \hat{d}_i + \hat{d}_1) \cdot sat\left(e_{i-1}^j, \lambda\right), \\ &i = 1, 2, \cdots, N-1 \end{aligned} \tag{57}$$

Note 2: Theorems 5 and 6 are valid despite the uncertainty and disturbance, because the nature of their proof does not depend on the existence or non-existence of uncertainty and disturbance. Therefore, the problem of circular multi-mode synchronization is addressed in the presence of uncertainty and disturbance.

## 3. Application in Secure Communication Based on Mapping and Chaotic Masking

Chaotic signals have complex behavior and are thus hardly predictable. Using chaotic signals as message carriers in secure communications and cryptography is a suitable and secure solution. Chaotic masking is a way to hide the signal (integration of synchronization with masking) [33,37]. In this approach, the message signal is added to a linear combination of state vector components of the master system. In other words, the message signal information is concealed within the chaotic behavior of the state components, which can enhance communication channel security. The message signal can be recovered from the receiver side by synchronizing the slave system with the master system. Ensuring the synchronization error convergence to zero and the disturbance signals and unknown parameters in the master and slave systems can increase security in the communication channel.

The following lines discuss the application of the proposed method in secure communication for chaotic masking. Suppose $m(t)$ is a message. It is encrypted with a proper map:

$$m_0(t) = \Lambda(m(t), f(t), a) \tag{58}$$

where $\Lambda(m(t), f(t), a)$ is a definite and continuous function as the map, and $f(t)$ is a definite and continuous signal as a coder. For instance, we can define $\Lambda(m(t), f(t), a)$ as follows:

$$\Lambda(m(t), f(t), a) = \tanh(a \cdot m(t) + f(t)),$$
$$f(t) = 0.2\sin(10t) + 0.1\sin(20\pi t) + 0.05\cos(2\pi t), a \in R \tag{59}$$

where $a$ is a coefficient so that $|a \cdot m(t) + f(t)| \leq 4$ will stand.

Signals $m_0(t)$, $f(t)$ will be masked as follows and transmitted in two components different from the chaotic system:

$$\widetilde{m}(t) = m_0(t) + \sum_{i=1}^{n} \lambda_i x_i \quad \widetilde{f}(t) = f(t) + \sum_{i=1}^{n} \mu_i x_i \tag{60}$$

The receiver initially obtains the estimation of signals $m_0(t)$ and $f(t)$. Then, it will calculate $\hat{m}_0(t)$, and, ultimately, we will calculate $\hat{m}(t)$ as follows:

$$\hat{m}_0(t) = \widetilde{m}(t) - \sum_{i=1}^{n} \lambda_i y_i = $$
$$m_0(t) + \sum_{i=1}^{n} \lambda_i x_i - \sum_{i=1}^{n} \lambda_i y_i = m_0(t) + \sum_{i=1}^{n} \lambda_i e_i \rightarrow m_0(t) \tag{61}$$

$$\hat{f}(t) = \widetilde{f}(t) - \sum_{i=1}^{n} \mu_i y_i = f(t) + \sum_{i=1}^{n} \mu_i x_i - \sum_{i=1}^{n} \mu_i y_i = f(t) + \sum_{i=1}^{n} \mu_i e_i \rightarrow f(t) \tag{62}$$

To recover the signal of message $m(t)$, we can do as follows:

$$m_0(t) = \Lambda(m(t), f(t), a) \rightarrow \hat{m}_0(t) = \Lambda\left(\hat{m}(t), \hat{f}(t), a\right) = \tanh\left(a \cdot \hat{m}(t) + \hat{f}(t)\right)$$
$$\Rightarrow \hat{m}(t) = \frac{1}{a}\left(tanh^{-1}\left(\hat{m}_0(t) - \hat{f}(t)\right)\right) \tag{63}$$

Figure 3 shows the chaotic masking in multi-mode synchronization.

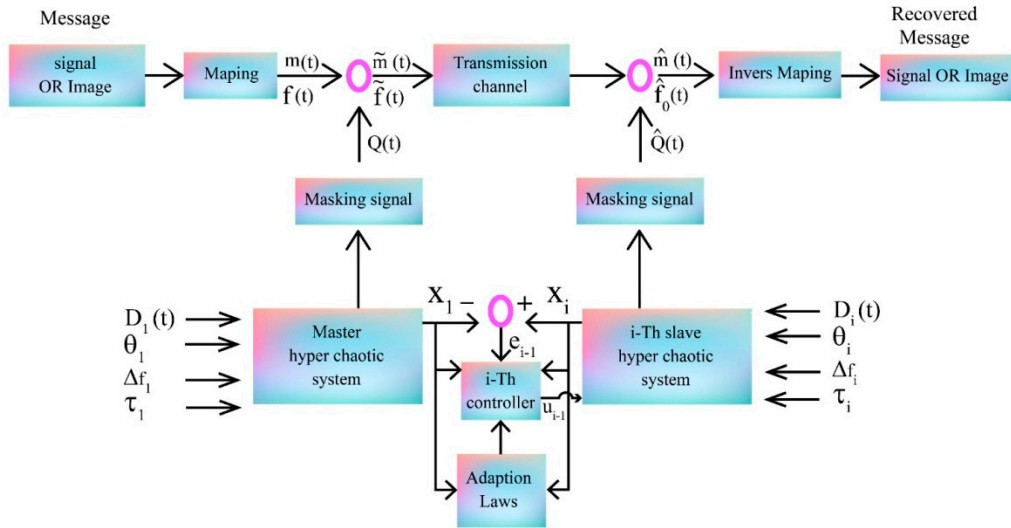

**Figure 3.** Block diagram of chaotic masking with multi-state synchronization.

Various statistical methods demonstrate the efficiency of synchronization systems of chaotic systems using image encryption. The histogram, correlation, number of pixels change rate (NPCR), unified average changing intensity (UACI), peak signal-to-noise ratio (PSNR), and information entropy parameters were used to demonstrate the efficacy of the proposed synchronization method for the synchronization of fractional-order chaotic systems. The description of the image evaluation parameters is given in references [45–49].

## 4. Simulation and Results

Here, the fractional-order hyper-chaotic Chen system [47] and two fractional-order Chen systems are considered as master and slave, respectively. They were determined as follows:

$$
\begin{cases}
D^q x_{11} = \theta_{11}(x_{12} - x_{11}) + x_{14}(t - \tau_1) \\
D^q x_{12} = \theta_{21} x_{11} - x_{11} x_{13} + \theta_{13} x_{12} \\
D^q x_{13} = x_{11} x_{12} - \theta_{14} x_{13} \\
D^q x_{14} = x_{12} x_{13} + \theta_{15} x_{14}
\end{cases}
\tag{64}
$$

$$
\begin{cases}
D^q x_{31} = \theta_{31}(x_{32} - x_{31}) + x_{34}(t - \tau_3) + u_{21} \\
D^q x_{32} = \theta_{31} x_{31} - x_{31} x_{33} + \theta_{33} x_{32} + u_{22} \\
D^q x_{33} = x_{31} x_{32} - \theta_{34} x_{33} + u_{23} \\
D^q x_{34} = x_{32} x_{33} + \theta_{35} x_{34} + u_{24}
\end{cases}
\quad
\begin{cases}
D^q x_{21} = \theta_{21}(x_{22} - x_{21}) + x_{24}(t - \tau_2) + u_{11} \\
D^q x_{22} = \theta_{22} x_{11} - x_{21} x_{23} + \theta_{23} x_{22} + u_{12} \\
D^q x_{23} = x_{21} x_{22} - \theta_{24} x_{13} + u_{13} \\
D^q x_{24} = x_{22} x_{13} + \theta_{25} x_{14} + u_{14}
\end{cases}
\tag{65}
$$

where $\theta_{11}$, $\theta_{12}$, $\theta_{13}$, $\theta_{14}$, $\theta_{15}$ are unknown parameters. When $\theta_{i4} = 0.3$, $\theta_{i1} = 35$, $\theta_{i2} = 3$, $\theta_{i3} = 12$, $\theta_{i5} = 7$, $i = 1, 2, 3$, then systems (64) and (65) are chaotic systems. The expressions:

$$
u_1 = [u_{11}, u_{12}, u_{13}, u_{14}]^T, u_2 = [u_{21}, u_{22}, u_{23}, u_{24}]^T
$$

are control inputs and

$$
f_1(x_1) = \begin{bmatrix} 0 \\ -x_{11}x_{13} \\ x_{11}x_{12} \\ x_{12}x_{13} \end{bmatrix}, \quad
F_1(x_1) = \begin{bmatrix} (x_{12} - x_{11}) \\ x_{11} \\ x_{12} \\ x_{14} \end{bmatrix}
$$

$$
f_3(x_3) = \begin{bmatrix} 0 \\ -x_{31}x_{33} \\ x_{31}x_{32} \\ x_{32}x_{33} \end{bmatrix}, \quad
F_3(x_3) = \begin{bmatrix} (x_{32} - x_{31}) \\ x_{31} \\ x_{32} \\ x_{34} \end{bmatrix}, \quad
f_2(x_2) = \begin{bmatrix} 0 \\ -x_{21}x_{23} \\ x_{21}x_{22} \\ x_{22}x_{23} \end{bmatrix}, \quad
F_2(x_2) = \begin{bmatrix} (x_{22} - x_{31}) \\ x_{21} \\ x_{22} \\ x_{24} \end{bmatrix}
$$

Error dynamics systems are attained as follows:

$$
\begin{cases}
D^q e_{11} = \theta_{21}(x_{22} - x_{21}) - \theta_{11}(x_{12} - x_{11}) + x_{24}(t - \tau_2) - x_{14}(t - \tau_1) + u_{11} \\
D^q e_{12} = (\theta_{22} x_{21} - x_{21} x_{23} + \theta_{23} x_{23}) + (-\theta_{12} x_{11} + x_{11} x_{13} - \theta_{13} x_{31}) + u_{12} \\
D^q e_{13} = (x_{21} x_{22} - \theta_{24} x_{23}) + (-x_{11} x_{12} + \theta_{14} x_{13}) + u_{13} \\
D^q e_{14} = (x_{22} x_{13} + \theta_2 x_{24}) + (-x_{12} x_{13} - \theta_{15} x_{14}) + u_{14}
\end{cases}
\tag{66}
$$

$$
\begin{cases}
D^q e_{21} = \theta_{31}(x_{32} - x_{31}) - \theta_{11}(x_{12} - x_{11}) + x_{34}(t - \tau_2) - x_{14}(t - \tau_1) + u_{21} \\
D^q e_{21} = (\theta_{32} x_{31} - x_{31} x_{33} + \theta_{33} x_{33}) + (-\theta_{12} x_{11} + x_{11} x_{13} - \theta_{13} x_{31}) + u_{22} \\
D^q e_{23} = (x_{31} x_{32} - \theta_{34} x_{33}) + (-x_{11} x_{12} + \theta_{14} x_{13}) + u_{23} \\
D^q e_{24} = (x_{32} x_{33} + \theta_{35} x_{34}) + (-x_{12} x_{13} - \theta_{15} x_{14}) + u_{24}
\end{cases}
\tag{67}
$$

In order to simulate, the initial states of the master system and two slave systems are chosen as follows:

$$x_1(0) = \begin{bmatrix} 10 \\ 10 \\ 10 \\ 10 \end{bmatrix} x_2(0) = \begin{bmatrix} 2 \\ 2 \\ 2 \\ 2 \end{bmatrix} x_3(0) = \begin{bmatrix} 3 \\ 3 \\ 3 \\ 3 \end{bmatrix}$$

It is assumed that the initial values of the adaptive parameters are as follows:

$$\hat{\theta}_1(0) = \begin{bmatrix} 3 \\ 3 \\ 3 \\ 3 \end{bmatrix} \hat{\theta}_2(0) = \begin{bmatrix} 3 \\ 3 \\ 3 \\ 3 \end{bmatrix} \hat{\theta}_3(0) = \begin{bmatrix} 4 \\ 4 \\ 4 \\ 4 \end{bmatrix} \sigma_i = 10. \ i = 1, 2, 3, 4, 5$$

The parameters of the controller gains $K_{i-1}$ are as follows:

$$K_1 = \begin{bmatrix} -12 & 0 & 0 & 0 \\ 0 & -12 & 0 & 0 \\ 0 & 0 & -12 & 0 \\ 0 & 0 & 0 & -12 \end{bmatrix}, K_2 = \begin{bmatrix} -12 & 0 & 0 & 0 \\ 0 & -12 & 0 & 0 \\ 0 & 0 & -12 & 0 \\ 0 & 0 & 0 & -12 \end{bmatrix}$$

Time-varying delay for master and slave systems has been considered as below:

$$\tau_1(t) = \begin{cases} 1 & 0 \leq t \leq 2 \\ 3 & 2 \leq t \leq 4 \\ 1 & t \geq 4 \end{cases} \quad \tau_2(t) = \begin{cases} 1 & 0 \leq t \leq 3 \\ 2 & 3 \leq t \leq 6.5 \\ 1 & t \geq 6.5 \end{cases} \quad \tau_3(t) = \begin{cases} 1 & 0 \leq t \leq 2.5 \\ 3 & 2.5 \leq t \leq 5.5 \\ 1 & t \geq 5.5 \end{cases}$$

Later, phase diagrams for chaotic synchronization systems will be depicted. Figure 4 illustrates the chaotic behavior of the fractional-order system. Master and slave systems display chaotic behavior, and their parameters are determined so that the behavior of all the systems will be chaotic.

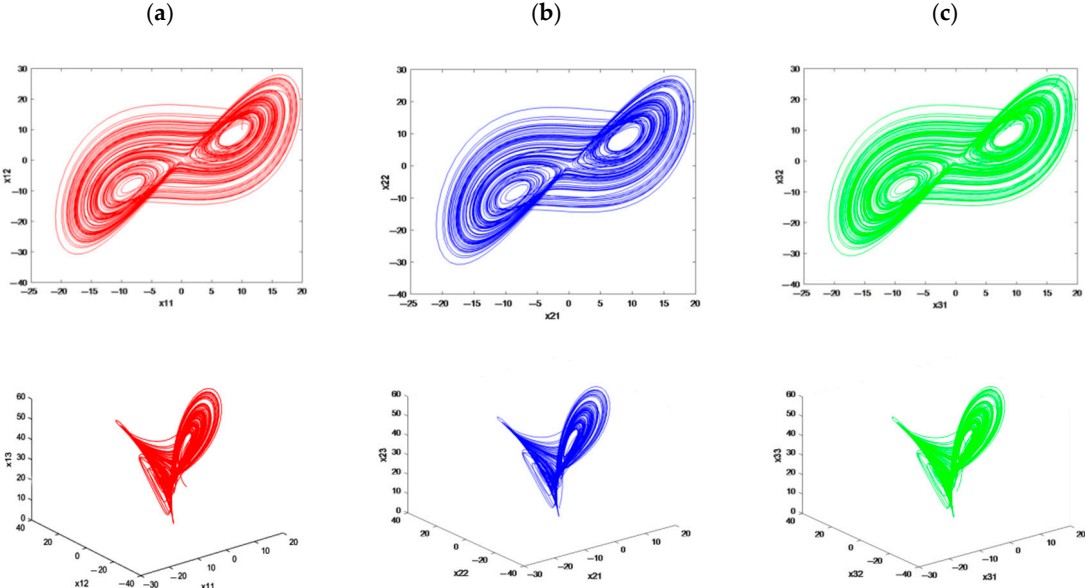

**Figure 4.** Phase diagrams for master and slave systems. (**a**) Master System, (**b**) slave System 1, (**c**) slave system 2.

As can be seen from Figure 4, the phase curves of all three systems are chaotic.

In the simulations, it is supposed that the main parameters vary with time stepwise. The values of disturbances and uncertainties are supposed as follows:

$$\Delta f_1 = \begin{bmatrix} 0.01 x_{11}^2 \cos(x_{11}) \\ 0.02 \sin(x_{12}) \\ 0.015 x_{13} \cos(x_{13}) \\ 0.025 \sin(x_{14}) \end{bmatrix} \cdot g_1(x_1) = |x_1|^2 + 20|x_1|$$

$$\Delta f_2 = \begin{bmatrix} 0.15 \sin(x_{21}) \\ 0.025 x_{22} \cos(x_{22}) \\ 0.035 \sin(x_{23}) \\ 0.02 x_{24} \sin(x_{24}) \end{bmatrix} \cdot g_2(x_2) = |x_2|$$

$$\Delta f_3 = \begin{bmatrix} 0.025x_{31}\sin(x_{31}) \\ 0.015x_{31}x_{32}\sin(x_{32}) \\ 0.035\sin(x_{33}) \\ 0.02x_{34}\sin(x_{34}) \end{bmatrix} \cdot g_3(x_3) = |x_3|^2 + 14|x_3|$$

$$D_1 = \begin{bmatrix} 0.025\sin\left(\frac{\pi}{3}t\right) \\ 0.05\sin\left(\frac{\pi}{3}t\right) \\ 0.05\sin\left(\frac{\pi}{6}t\right) \\ 0.04\sin\left(\frac{\pi}{5}t\right) \end{bmatrix} \quad D_2 = \begin{bmatrix} 0.035\sin\left(\frac{\pi}{3}t\right) \\ 0.045\sin\left(\frac{\pi}{3}t\right) \\ 0.025\sin\left(\frac{\pi}{5}t\right) \\ 0.03\sin\left(\frac{\pi}{6}t\right) \end{bmatrix} \quad D_3 = \begin{bmatrix} 0.015\sin\left(\frac{\pi}{3}t\right) \\ 0.025\sin\left(\frac{\pi}{2}t\right) \\ 0.035\sin\left(\frac{\pi}{10}t\right) \\ 0.01\sin\left(\frac{\pi}{25}t\right) \end{bmatrix}$$

The estimation errors of the system parameters and time delay errors are shown in Figure 5. The paths of error dynamics systems and control efforts are exhibited in Figure 6.

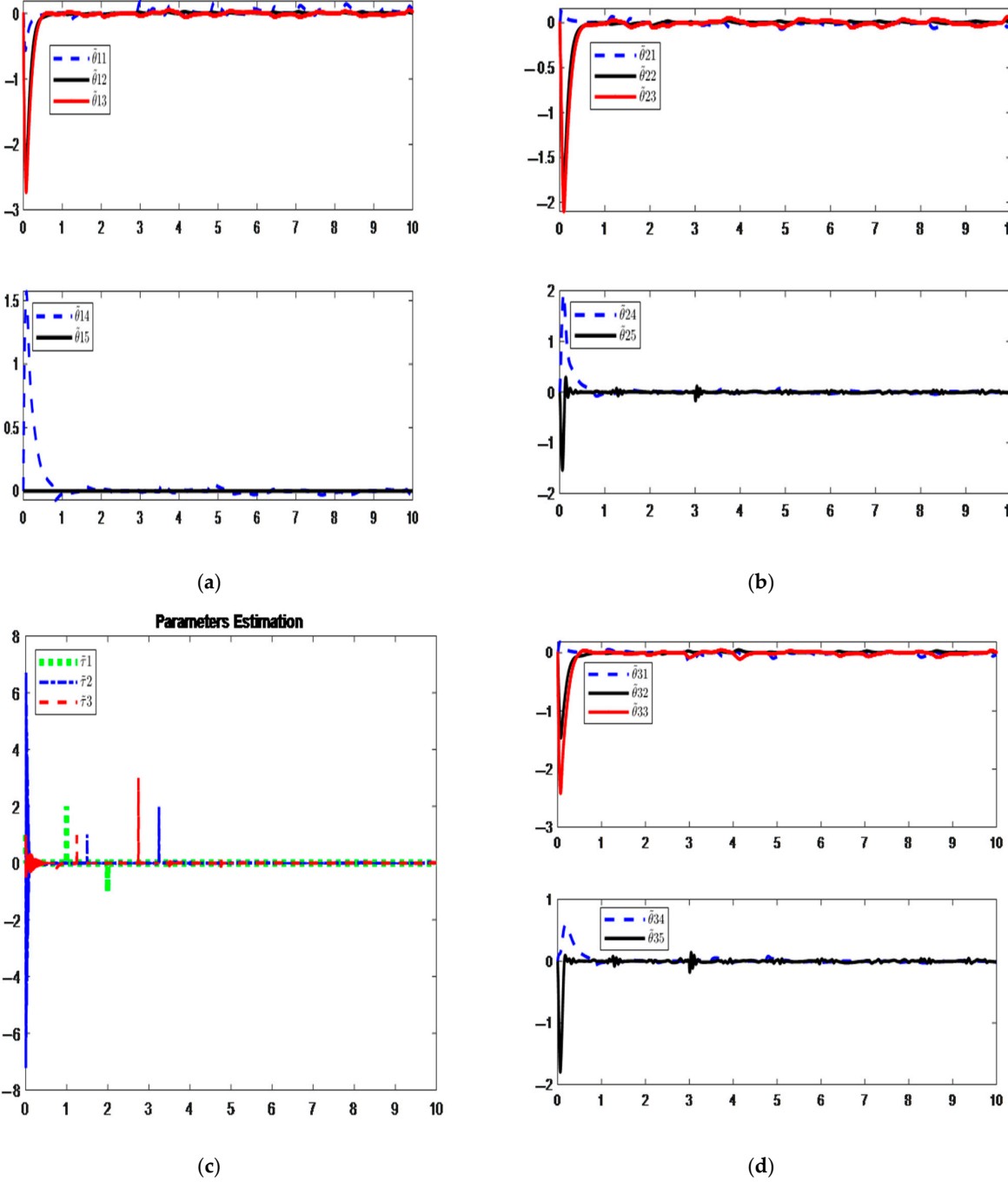

(a)

(b)

(c)

(d)

**Figure 5.** Estimation of parameters and time delay errors in multi-mode synchronization obtained during disturbance time delay and uncertainty. ((**c**) indicates time delay errors and (**a**,**b**,**d**) indicate estimation of errors uncertainty).

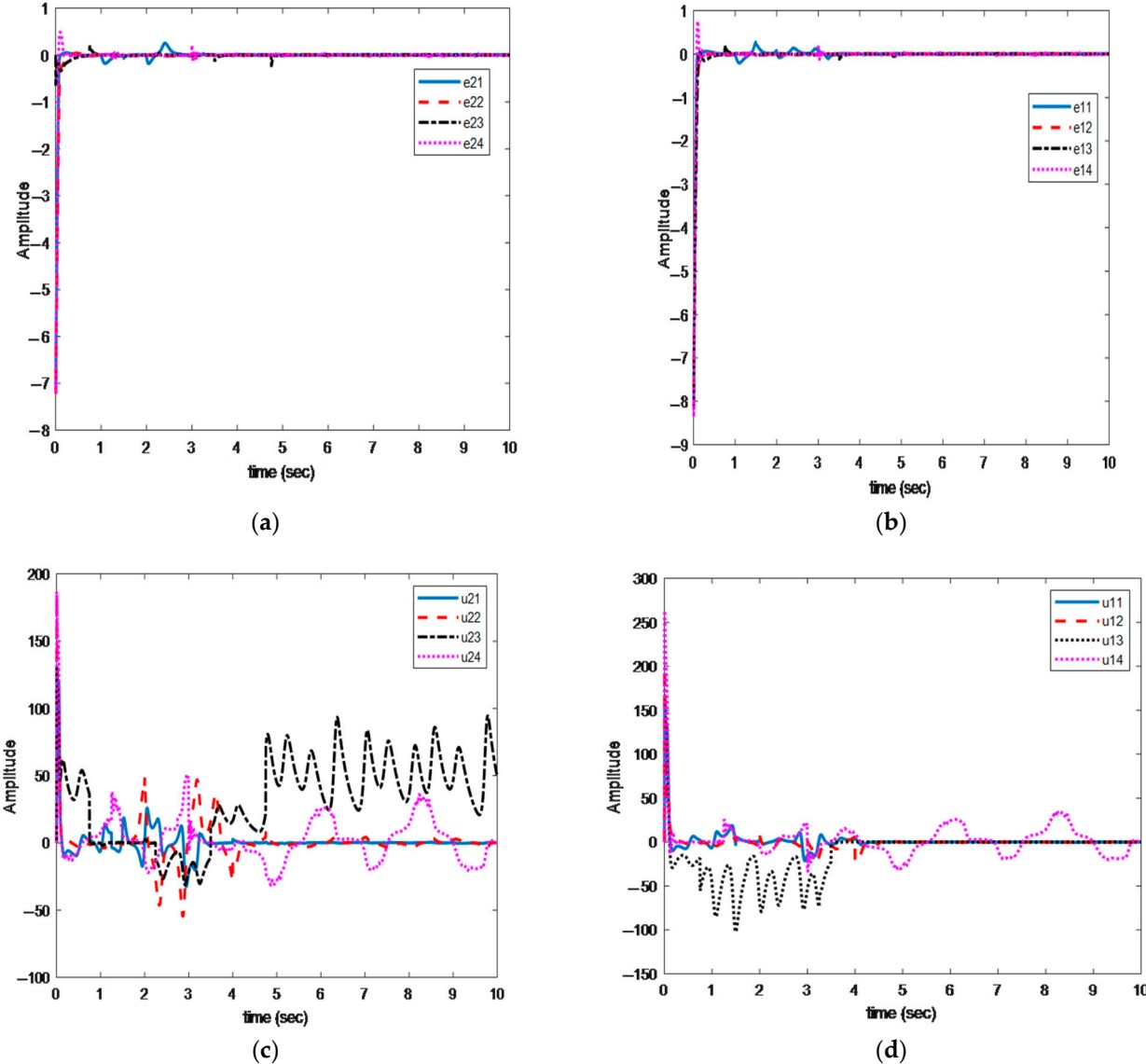

**Figure 6.** Curves of synchronization errors obtained during disturbance and uncertainty. (subfigures (**a**,**b**) show dynamic state errors and subfigures (**c**,**d**) show control efforts).

Figure 5 shows that the parameters and time delay errors in the multi-mode synchronization properly converge to zero despite the uncertainty and disturbance. The figure shows minimal fluctuation. Therefore, the synchronization approach produced successful results despite the parametric changes. Figure 5 depicts the multi-mode synchronization errors during disturbance and structural uncertainty. Figure 6 presents the error values for the disturbance and uncertainty boundaries estimation.

As indicated in Figure 6,despite the disturbance and uncertainty, the errors were large initially and then approached zero with very little fluctuation. There were fluctuations in these control efforts. Synchronization errors altered slightly with parameter variation, which rapidly converged to zero over time.

Figure 7 indicates that the errors in estimating the uncertain boundaries were initially large due to the initial values of the parameters but are approximated to zero over time. Additionally, the errors related to the uncertainty have converged with very little oscillation to near-zero values. The proper and fast estimation of error boundaries as well as the rapid identification of system parameters leads to proper controller performance. Overall, despite the disturbances, uncertainties, and parametric changes in the master and slave hyper-chaotic systems, the multi-mode synchronization, with the help of the proposed adaptive controller, performed desirably and produced excellent results.

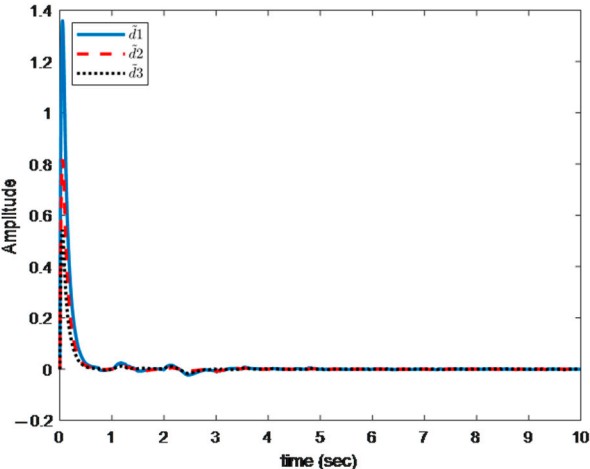 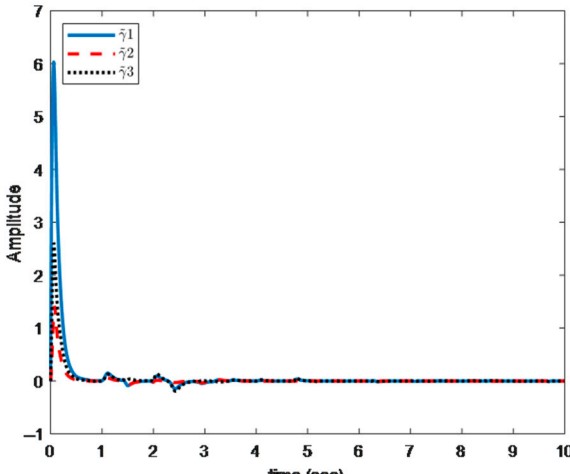

**Figure 7.** Error curves obtained for estimating the uncertain boundaries (**right**) and disturbances (**left**).

## 5. Experiment Results

This section describes the results of the synchronization method of the fractional-order Chen chaotic systems for image encryption. First, the images used to perform the experiments were investigated. In the next stage, the encrypted images were represented using the synchronization methods of the fractional-order Chen chaotic systems for $q = 0.97$, followed by illustrating the histogram of the encrypted images for $q = 0.97$. Finally, various statistical parameters indicated the effectiveness of the synchronization methods with the fractional-order chaotic system.

Figure 8 illustrates the results obtained from the encryption and decryption results of the proposed secure communication design in five benchmark images.

Figure 8 shows the proper encryption and decryption of the five benchmarks. Figure 8 gives a more precise evaluation by comparing the histograms of the main and recovered images.

Figure 9 shows the main images on the left. In the next column, the histogram of the encrypted image is illustrated. The third column depicts the recovered image. The fourth column shows the histogram of the recovered image. This figure reveals that the histograms of the main image and recovered images have slight differences. However, for more precise analysis, Table 1 demonstrates the results of various statistical criteria. This table shows histogram criteria, correlation, UACI, NPCR, PSNR, and information entropy.

| **Input Image** | **Encoded** | **Decoded** |
| :---: | :---: | :---: |

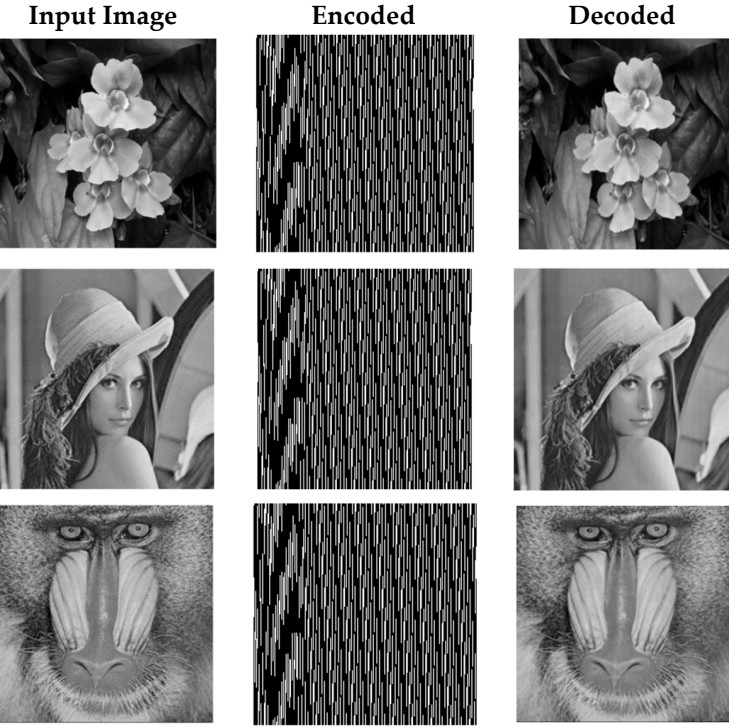

**Figure 8.** *Cont.*

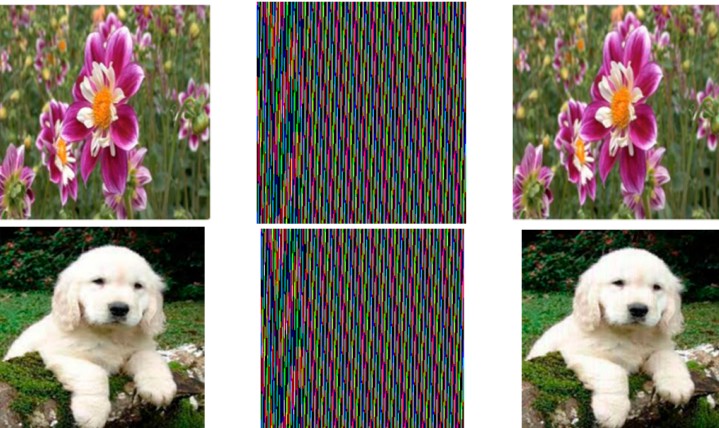

**Figure 8.** Displayed Benchmark images encryption using synchronization of the fractional-order Chen systems (*q* = 0.97).

| Input Images | Histogram Input Image | Decoded Image | Histogram Encoded Image |
|---|---|---|---|

**Figure 9.** *Cont.*

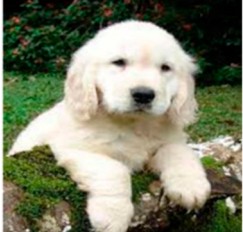 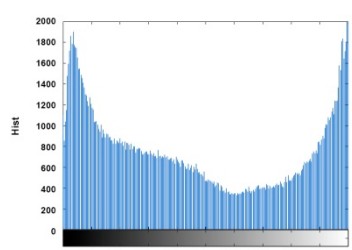 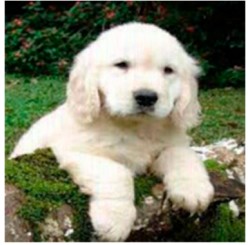 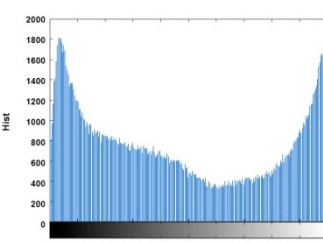

**Figure 9.** Displayed histograms for various Benchmark images encrypted using synchronization of the fractional-order chaotic system ($q = 0.97$).

**Table 1.** Statistical metrics Benchmark images ($q = 0.97$).

| Images | Histogram | | | Correlation | Differential Attack | | PSNR | Information Entropy |
| | Main | Encrypted | Decoded | | NPCR (%) | UACI (%) | | |
|---|---|---|---|---|---|---|---|---|
| **Image 1** | 36,233.046875 | 10,278,639.63281 | 35,454.91406 | 0.9996 | 99.99 | 33.46 | 43.1049 | 7.590 |
| **Image 2** | 40,953.960938 | 10,271,802.38281 | 40,603.99218 | 0.9993 | 99.98 | 33.55 | 42.8879 | 7.435 |
| **Image 3** | 61,975.74210 | 10,270,160.96093 | 70,547.03900 | 0.9989 | 99.69 | 33.16 | 43.0529 | 7.210 |
| **Image 4** | 75,071.242188 | 3,805,961.023438 | 75,522.007813 | 0.9996 | 99.61 | 33.46 | 45.7973 | 7.421 |
| **Image 5** | 17,756.507813 | 3,806,198.531250 | 17,534.671875 | 0.9998 | 99.92 | 33.47 | 43.9077 | 7.822 |

Table 1 reveals that the histograms obtained from the main and the recovered images have slight differences. The correlation between the main and the recovered images is extremely close to 1. The NPCR is approximately 100%. Moreover, good values were obtained for UACI and PSNR. All five images have an information entropy greater than 7. The results suggest that the proposed communication design method successfully encrypted and recovered the main images.

## 6. Discussion, Advantages and Disadvantages, and Future Works

This paper presents a new secure communication based on Chen fractional order-order chaotic synchronization. This study mainly aims to propose a synchronization method based on multi-mode adaptive control and the Chen fraction order chaos system to encode images. In this work, experiments on five benchmark images were used with $256 \times 256$ dimensions. Next, the cryptographic method based on synchronization of the Chen fractional order chaos system was applied to benchmark images. The fractional order Chen chaotic system had different delay and q values in this section. Moreover, the presence of an unknown delay factor in the systems complicated the synchronization problem.

Next, the masking technique of chaotic systems was used to encode images. Then, the parameters of histogram, correlation, N.P.C.R., U.A.C.I., P.S.N.R., and information entropy for benchmark images were obtained to demonstrate the efficiency of the proposed synchronization method. The experimental results indicated that the Chen fractional order synchronization method performed successfully. Derivative change varies the behavior of the system as a whole. This is an essential point in cryptography that reduces the possibility of decryption. Considering that the image data are combined with chaotic signals, the amount of security in encryption is sufficiently high. In this mechanism, the parameters of system and time delays are unknown. Moreover, the presence of nonlinear uncertainty makes its detection harder. Results from the rigorous security analysis (fractional order, correlation coefficient, entropy, NPCR, etc.) proved that the proposed method was robust and that the proposed scheme had a high security order. Thus, it can be used for real-time transmission of other images.

Table 2 indicates the high efficiency of the proposed method compared to other studies.

According to Table 2, it can be perceived that our proposed method has a high efficiency compared to other research.

Reviewing the previous studies indicates that little attention has been paid to time delays in communication designs with fractional chaotic system synchronization. The available secure communication synchronization designs ignore unknown time delays, exogenous disturbances, and unknown parameters at the same time. However, our proposed design is comprehensive, considers all factors, and is accompanied by a new function introduced for secure communication based on a flexible map with time-varying coding. The simulations showed that the proposed algorithm performed flawlessly against parameter changes, disturbance, time delays, and uncertainty. The guarantee of system stability is based on Lyapunov's function. The main reason for the proper performance of the proposed method lies in the form of updated rules obtained in the article; these rules were defined in such a way that the necessary and sufficient conditions for stability in the concept of Mittag-Leffler were established. On the other hand, the method can quickly estimate the parameters and the error of estimating the disturbance and uncertainty boundaries, which results in a faster reaction of the controller against undesirable factors. In general, the proposed method meets all the requirements and has the ability to be implemented in different applications.

**Table 2.** Comparison of the proposed method with other related works.

| Reference | Dataset | Types of Data | Encryption Method | Details Encryption Method | Type of Delay | Details of Method | |
|---|---|---|---|---|---|---|---|
| | | | | | | Unknown Parameters | Disturbance |
| [30] | sinusoids signal | - | Fractional-Order Chaotic Systems | Genesio–Tesi system | × | × | √ |
| [31] | Voltage signal | - | Fractional-Order Chaotic Systems | Seven-dimensional fractional-order chaotic system | × | × | × |
| [32] | sinusoids signal | - | Fractional-Order Chaotic Systems | Exponential Chaotic System | × | × | × |
| [33] | sinusoids signal | - | Fractional-Order Chaotic Systems | FO complex chaotic Lü system | × | × | × |
| [34] | sinusoids signal | - | Fractional-Order Chaotic Systems | A Novel Fractional Order Chaotic System | × | × | × |
| [35] | square signal | - | fractional-order chaotic systems | Chua system | × | × | × |
| [36] | sinusoids signal | - | fractional-order chaotic systems | Fractional chaotic T system via matrix projective | × | √ | √ |
| [37] | sinusoids signal | - | fractional-order chaotic systems | fractional-order Chen and lu hyper-chaotic systems | × | √ | × |
| [39] | Benchmark and Medical Image | Color images | Integer-order chaotic systems | Fast Reaching Finite Time synchronization | × | × | √ |
| [50] | Benchmark Images | Color images | Fractional order system | Fractional Order Chaotic Systems | × | × | × |
| [51] | Benchmark Images | Gray Scale Images | Fractional order system | Fractional-Order Simplest Chaotic | × | × | × |
| [52] | - | - | - | - | Constant-known | √ | × |
| [53] | - | - | - | - | Constant-known | × | × |
| [54] | - | sine signal | - | chaotic masking | Time varying-known | √ | √ |
| [55] | "Travelling" music in Matlab | speech signal | complex Lü systems | self-time-delay synchronization and chaotic masking | constant-known | × | × |
| **Proposed method** | Benchmarks Images | Gray Scale & color Images | Fractional order system | Multi-Mode Synchronization of Fractional-Order Chaotic Systems | Time varying-unknown | √ | √ |

In the proposed method, four significant and real factors, including disturbance, uncertainty, time delays, and parameter changes, are considered simultaneously, which helps achieve the highest performance. It is also beneficial in practical implementations. Still, the presented approach applies to a variety of chaotic systems. The resulting control law is considered as an explicit and continuous function. Additionally, the proposed theorems have the ability of two types of multi-mode synchronization. Additionally, the amplitude of the control function is generally significant in the proposed method; if the changes in the parameters are permanent and at a high rate, the recommended algorithm will not yield outstanding performance.

Being based on the Caputo fractional-order derivative can be considered one of the disadvantages of this method. Moreover, parameter changes are only considered as a step change. Proving equivalence between the two types of synchronization, proving convergence of all the errors to zero in the presence of disturbance and uncertainty, the continuation of control laws, and proving efficiency are some of the advantages of this proposed method. To develop the proposed method and solve its weaknesses, we must do as follows: develop the proposed method for other fractional-order derivatives, develop the proposed method for parameters with permanent changes, and analyze the application of the proposed method in secure communication.

It is recommended that future research should address the values of time-varying fractional orders, since reference [56] suggests that this could describe the system more effectively. As future tasks in the synchronization method section, first-and-second-order fuzzy controllers can be adopted. Due to the uncertainty in medical data, fuzzy systems (type 1 and type 2) are highly productive. Modeling chaotic systems using fuzzy systems and fuzzy controller design can provide better answers to the synchronization problem. Applying a fuzzy polynomial system is highly effective in chaotic systems modeling and in their synchronization.

## 7. Conclusions

This paper investigated the multi-mode synchronization of fractional-order hyper-chaotic systems in the presence of uncertainty, disturbance, time delays, and parameters with stepwise changes in both transmission and circular modes. First, the results confirmed that they are equivalent to the synchronization method. Then, by defining the appropriate Lyapunov function and proofing the theorems, the rules for updating the parameters, as well as error estimating the disturbance and uncertainty boundaries, were assigned. Nonetheless, it should be noted that the stability of the dynamical synchronization system guarantees that all of the synchronization errors, the parameter estimation errors, the disturbance boundary estimation errors, and the uncertainty boundary estimation errors converge to zero. Determining the control law as an explicit and continuous function inhibited chattering. The results also indicated that the rate of change of parameters was not high since the proposed algorithm still displayed an acceptable performance. Simulations were conducted in the presence and absence of disturbance and uncertainty to investigate the effectiveness of the proposed method. The proposed controller reduced synchronization errors, disturbance errors, and uncertainties to zero with low fluctuation near zero, despite changes in time-varying parameters of hyper-chaotic systems. During the parameters changing, a small deviation in synchronization errors was quickly depreciated.

Moreover, unknown time delays were considered. With the help of adaptive regulations, these delays were estimated and used in equations of control effort. Despite these delays, the results revealed that synchronization was appropriately conducted. Additionally, a novel masking design was proposed using the hyperbolic tangent modulation function. To assess this design, benchmark images were encrypted. Various statistical criteria including PSNR, UACI, NPCR, histogram, and correlation were analyzed. The simulation results indicated that the proposed design was successful. The obtained higher efficiency results clearly demonstrated the superiority of the proposed method.

**Author Contributions:** Conceptualization, A.A.K.J., R.A. and A.Z.; methodology, A.A.K.J., R.A. and A.Z.; software, A.A.K.J. and A.Z.; validation, A.A.K.J., R.A. and A.Z.; formal analysis, A.A.K.J. and S.B.; investigation, A.A.K.J., R.A. and A.Z.; resources, A.A.K.J. and A.Z.; data curation, A.A.K.J. and A.Z.; writing—original draft preparation, A.A.K.J., R.A. and A.Z.; writing—review and editing, A.A.K.J. and A.Z.; visualization, A.A.K.J., R.A., S.B. and A.Z.; supervision, A.Z. All authors have read and agreed to the published version of the manuscript.

**Funding:** This research received no external funding.

**Institutional Review Board Statement:** Not applicable.

**Informed Consent Statement:** Not applicable.

**Data Availability Statement:** Not applicable.

**Conflicts of Interest:** The authors declare no conflict of interest.

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
