# Peer review of "Robust Multi-Mode Synchronization of Chaotic Fractional Order Systems in the Presence of Disturbance, Time Delay and Uncertainty with Application in Secure Communications"

_2504-2289, doi:10.3390/bdcc6020051_

Round 1

Reviewer 1 Report

My concerns have been addressed.

Author Response

Thanks a lot for your comments.

We did our best to modify the text of the manuscript.

Reviewer 2 Report

In my opinion, the manuscript may be accepted for publication as it is. All of the review comments are fixed.

Author Response

Thanks a lot for your comments.

We did our best to modify the text of the manuscript.

This manuscript is a resubmission of an earlier submission. The following is a list of the peer review reports and author responses from that submission.

Round 1

Reviewer 1 Report

The work presented the robust multi-mode synchronization of chaotic fractional order systems in the presence of disturbance, time delay and uncertainty with application in gray scale images encryption. The analysis is complete and the study seems to be solid. Some comments are given as follows:

1) The abstract should be more concise and refined.

2) Is the encryption algorithm suitable color images? I suggest the authors to add some corresponding results.

3) The use of the synchronization in image encryption is not clear. It is just work on the design of chaotic image encryption algorithm. Please give some explanations.

4) The advantages of the presented synchronization control method should be illustrated.

5) Some recent related work on chaos synchronization and chaotic image encryption can be introduced to highlight the difference and novelty of the work from some existing work, such as Design and analysis of multiscroll memristive Hopfield neural network with adjustable memductance and application to image encryption. IEEE Transactions on Neural Networks and Learning Systems, 2022, DOI: 10.1109/TNNLS.2022.3146570; Hidden coexisting hyperchaos of new memristive neuron model and its application in image encryption. Chaos, Solitons & Fractals, 2022, 158, 112017; Dynamic analysis, circuit realization, control design and image encryption application of an extended Lü system with coexisting attractors. Chaos, Solitons & Fractals, 2018, 114, 230-245, etc.

6) Some errors in English writing and references should be carefully corrected.

Reviewer 2 Report

The authors try to the design of an adaptive and robust synchronization of MMFOCS.

Unprofessional attitude to writing the manuscript, a lot of typos, missing experiments ( math quantity of histograms, average pixel intensity, entropy, entropy comparison, corr coeff analysis(H V D), NPCR, UACI, ...........).

Very poor values in Table 1. Information ENTROPY, not information theory. And etc.. Mix of poor paragraphs. I do not understand Table 2. Poor quality.